# To be Robust or to be Fair: Towards Fairness in Adversarial Training

## Abstract

Adversarial training algorithms have been proved to be reliable to improve machine learning models' robustness against adversarial examples. However, we find that adversarial training algorithms tend to introduce severe disparity of accuracy and robustness between different groups of data. For instance, PGD adversarially trained ResNet18 model on CIFAR-10 has 93% clean accuracy and 67% PGD $l_\infty$-8 adversarial accuracy[1] on the class "automobile" but only 59% and 17% on class "cat". This phenomenon happens in balanced datasets and does not exist in naturally trained models when only using clean samples. In this work, we theoretically show that this phenomenon can generally happen under adversarial training algorithms which minimize DNN models' robust errors. Motivated by these findings, we propose a Fair-Robust-Learning (FRL) framework to mitigate this unfairness problem when doing adversarial defenses and experimental results validate the effectiveness of FRL.

## 1 Introduction

The existence of adversarial examples (Goodfellow et al., 2014; Szegedy et al., 2013) causes huge concerns when applying deep neural networks on safety-critical tasks, such as autonomous driving vehicles and face identification (Morgulis et al., 2019; Sharif et al., 2016). These adversarial examples are artificially crafted samples which do not change the semantic meaning of the natural samples, but can misguide the model to give wrong predictions. As countermeasures against the attack from adversarial examples, adversarial training algorithms aim to train classifier that can classify the input samples correctly even when they are adversarially perturbed. Namely, they optimize the model to have minimum adversarial risk of that a sample can be perturbed to be wrongly classified:

$$\min_f \mathbb{E}_x \left[ \max_{||\delta|| \leq \epsilon} \mathcal{L}(f(x + \delta), y) \right]$$

These adversarial training methods (Kurakin et al., 2016; Madry et al., 2017; Zhang et al., 2019b) have been shown to be one type of the most effective and reliable ways to improve the model robustness against adversarial attacks. Although promising to improve model's robustness, recent studies show side-effects of adversarial training: it usually degrades model's clean accuracy (Tsipras et al., 2018).

In our work, we find a new intriguing property about adversarial training algorithms: they usually result in a large disparity of accuracy and robustness between different classes. As a preliminary study in Section 2, we apply natural training and PGD adversarial training (Madry et al., 2017) on the CIFAR10 dataset (Krizhevsky et al., 2009) using a ResNet18 (He et al., 2016) architecture. For a naturally trained model, the model performance in each class is similar. However, in the adversarially trained model, there is a severe performance discrepancy (both accuracy and robustness) of the model for data in different classes. For example, the model has high clean accuracy and robust accuracy (93% and 67% successful rate, separately) on the samples from the class "car", but much poorer performance on those "cat" images (59% and 17% successful rate). More preliminary results in Section 2 further show the similar "unfair" phenomenon from other datasets and models. Meanwhile, we find that this fairness issue does not appear in natural models which are trained on clean data. This fact demonstrates that adversarial training algorithms can indeed unequally help to improve model robustness for different data groups and unequally degrade their clean accuracy.

---

[1] The model's accuracy on the input samples that have been adversarially perturbed.

In this work we first define this problem as the *unfairness problem of adversarial training algorithms*. If this phenomenon happens in real-world applications, it can raise huge concerns about safety or even social ethics. Imagine that an adversarially trained traffic sign recognizer has overall high robustness, but it is very inaccurate and vulnerable to perturbations for some specific signs such as stop signs. The safety of this autonomous driving car is still not guaranteed. In such case, the safety of this recognizer depends on the worst class performance. Therefore, in addition to achieving overall performance, it is also essential to achieve fair accuracy and robustness among different classes, which can guarantee the worst performance. Meanwhile, this problem may also lead to the issues from social ethics perspectives, which are similar to traditional ML fairness problems (Buolamwini & Gebru, 2018). For example, a robustly trained face identification system might provide different qualitative levels of service safety for different ethnic communities.

In this paper, we first explore the potential reason which may cause this unfair accuracy / unfair robustness problem. In particular, we aim to answer the question - "*Will adversarial training algorithms naturally cause unfairness problems, such as the disparity of clean accuracy and adversarial robustness between different classes?*" To answer this question, we first propose a conceptual example under a mixture of two spherical Gaussian distributions which resembles to the previous work (Tsipras et al., 2018) but with different variances. In this setting, we hypothesize that adversarial training tends to only use robust features for model prediction, whose dimension is much lower than the non-robust feature space. In the lower dimensional space, an optimal linear model is more sensitive to the inherent data distributional difference and be biased when making predictions.

Motivated by these empirical and theoretical findings, we then propose a Fair Robust Learning (FRL) framework to mitigate this unfairness issue, which is inspired from the traditional debiasing strategy to solve a series of cost-sensitive classification problems but we make specific effort to achieve the fairness goal in adversarial setting. Our main contributions can be summarized as following: (a) We discover the phenomenon of "unfairness" problem of adversarial training algorithms and implement empirical studies to present this problem can be general; (b) We build a conceptual example to theoretically investigate the main reasons that cause this unfairness problem; and (c) We propose a Fair Robust Learning (FRL) framework to mitigate the unfairness issue in adversarial setting.

## 2 PRELIMINARY STUDIES

**CIFAR10** In this section, we present our preliminary studies to show that adversarial training algorithms usually present the unfairness issues, which are related to the strong disparity of clean accuracy and robustness among different classes. We implement algorithms including PGD adversarial training (Madry et al., 2017) and TRADES (Zhang et al., 2019b) on the CIFAR10 dataset (Krizhevsky et al., 2009). In CIFAR10, we both naturally and adversarially train ResNet18 (He & Garcia, 2009) models. In Figure 1, we present list the the model's accuracy and robustness performance (under PGD attack by intensity $4/255$ and $8/255$) for each individual class.

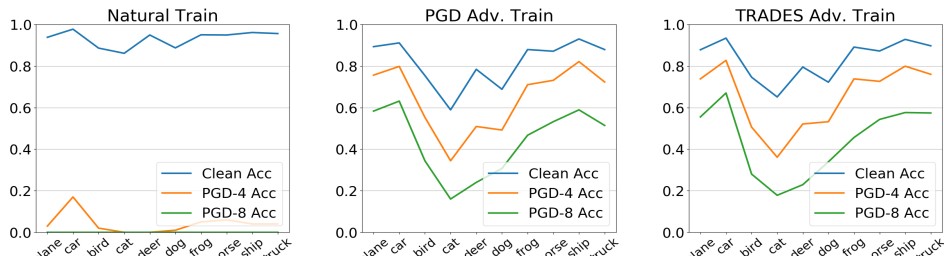

Figure 1: Clean and adversarial accuracy in each class of CIFAR10 dataset, from a naturally trained ResNet model (left), PGD-adversarially trained model (middle) and TRADES (right), against adversarial examples under $l_\infty$-norm by $8/255$. The trained models' robustness are evaluated by untargeted PGD attack under $l_\infty$-norm constrained by $8/255$ and $4/255$.

From the Figure 1, we can observe that – for the naturally trained models, every class has similar clean accuracy (around $90 \pm 5\%$) and adversarial accuracy (close to $0\%$) under the PGD attack. It suggests that naturally trained models do not have strong disparity of both clean and robustness performance among classes. However, for adversarially trained models (under PGD Adv. Training or TRADES), the disparity phenomenon becomes severe. For example, a PGD-adversarially trained model has $59.1\%$ clean accuracy and $17.4\%$ adversarial accuracy for the samples in the class "cat",

which are much lower than the model's overall performance. This phenomenon demonstrates that adversarial training algorithms cannot provide the same help for the robustness for the samples in class "cat" as other classes, and unfairly degrades too much clean accuracy for "cat". We list our empirical studies under more model architectures in Table 3 and more datasets (GTRSB (Stallkamp et al., 2011)) in Appendix A.2, where we can find the similar observations.

**GTSRB** We also investigate the fairness issue in German Traffic Sign Recognition Benchmark (GTRSB) (Stallkamp et al., 2011). It consists of 43 classes of images from different traffic signs, with image sizes $32 \times 32 \times 3$. In this dataset we also both naturally and adversarially train a 3-Layer CNN classifier. We list the model's performance and sort the classes in the order of decreasing clean accuracy and adv. accuracy. From the Figure 2, we can see that for the naturally trained model (left), most classes have high accuracy which is over 90%, but for adversarial training, some classes' accuracy drops by a large margin. Meanwhile, adversarial training also unequally improves the

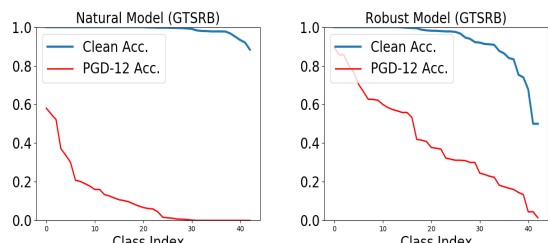

Figure 2: Class-wise Clean & Adversarial Accuracy on GTSRB of Naturally Trained Model (left) and Adversarially Trained Model (right).

model's robustness against PGD attacks given that some classes have very low adversarial accuracy. In this dataset, both natural model and robust model have clear distinguished adversairal accuracy (robustness) among classes.

## 3    THEORETICAL ANALYSIS BASED ON A CONCEPTUAL EXAMPLE

From our preliminary studies, we always observe that adversarially trained models have huge performance disparity (clean and adversarial accuracy) between different groups. In this section, we try to understand the unfairness problem via theoretical analysis based on a binary classification problem on a mixture-Gaussian distribution, which is similar to (Tsipras et al., 2018). We first state the necessary notions in this paper.

**Notations.** In the following, we use $f$ to denote the classification model which is a mapping $f :$ $\mathcal{X} \to \mathcal{Y}$ from input data space $\mathcal{X}$ and output labels $\mathcal{Y}$. Generally, naturally training will find the optimal $f$ to minimize the overall clean error $\mathcal{R}_{\text{nat}}(f) = \text{Pr.}(f(x) \neq y)$; and adversarially training will minimize the overall robust error $\mathcal{R}_{\text{rob}}(f) = \text{Pr.}(\exists \delta, ||\delta|| \leq \epsilon, \text{s.t.} f(x + \delta) \neq y)$. Specifically in the following binary classification problem, $\mathcal{Y} = \{-1, +1\}$ and each class's clean error and robust error are denoted as the conditional probabilities: $\mathcal{R}_{\text{nat}}(f, -1) = \text{Pr.}(f(x) = +1|y = -1)$, and $\mathcal{R}_{\text{rob}}(f, -1) = \text{Pr.}(\exists \delta, ||\delta|| \leq \epsilon, \text{s.t.} f(x + \delta) = +1|y = -1)$, respectively.

### 3.1    A BINARY CLASSIFICATION TASK

Our study is motivated by (Tsipras et al., 2018) which uncovers one key behavior of adversarial training: it excludes high-dimensional non-robust features (which are vulnerable to attack) and only preserves lower-dimensional robust features for prediction. Thus, in our case, we assume our conceptual dataset has the data-label pairs $(x, y)$ sampled from a distribution $\mathcal{D}$ follows:

$$y \overset{u.a.r}{\sim} \{-1, +1\}, \quad \theta = (\overbrace{\gamma, ..., \gamma}^{\text{dim} = m}, \overbrace{\eta, ..., \eta}^{\text{dim} = d}), \quad x \sim \begin{cases} \mathcal{N}(\theta, \sigma_{+1}^2 I) & \text{if } y = +1 \\ \mathcal{N}(-\theta, \sigma_{-1}^2 I) & \text{if } y = -1 \end{cases} \quad (1)$$

where $\mathcal{N}(\theta, \sigma_{+1}^2 I)$ is a normal distribution with mean vector $\theta$ and covariance matrix $\sigma_{+1}^2 I$ and same for class "-1". Following the work (Tsipras et al., 2018), we suppose that the feature space consists of two kinds of features: (a) **robust features** with center $\gamma$ and dimension $m$; and (b) **non-robust features** with center $\eta$ and dimension $d$. We assume $\eta < \epsilon < \gamma$, so an adversarial perturbation $\delta$ with intensity $||\delta||_\infty \leq \epsilon$ can manipulate a non-robust feature to have a different sign in expectation, but $\delta$ cannot attack a robust feature. Usually, the non-robust features' dimension $d$ is far higher than the robust features' dimension $d$, i.e., $(m << d)$.

In our case, we assume that the 2 classes have a key difference between their variances: $\sigma_{+1} : \sigma_{-1} = K : 1$, where $K > 1$. In this theoretical example, our main hypothesis is that: the variance difference between 2 classes will not lead to strong disparity of model performance for a naturally trained model whose prediction is based on a high dimensional feature space. However, the variance difference can cause large performance gap (both accuracy and robustness) for adversarially trained

models which are based on low-dimensional robust features. To illustrate this fact, we will explicitly calculate the 2 classes' clean and robust errors in the proposed distribution for both clean models and robust models.

## 3.2 Optimal Linear Model to Minimize Clean Error

We first calculate one linear model for this data to minimize the total clean errors. Specifically, we consider a linear classifier $f$ with its optimal parameters $w^*$ and $b^*$:

$$f^*(x) = \text{sign}(\langle w^*, x \rangle + b^*)$$
$$\text{where } w^*, b^* = \arg\min_{w,b} \Pr.(f(x) \neq y) \tag{2}$$

where $w$ is features' weight vector and $b$ is the model intersection. In later parts, we use $f$ to represent $f^*$ for convenience. We call the optimized model *naturally trained model* because it minimizes model's clean error to get overall high clean accuracy. Typically, a naturally trained model will use both robust features and non-robust features for inference but its prediction outcome majorly depends on non-robust features. Next, we show the exact form of the errors for each class in Theorem 1 and the proof is provided in Appendix B.1.

**Theorem 1** *Optimal linear classifier which minimizes overall clean error in $\mathcal{D}$ will have class-conditional clean errors:*

$$\mathcal{R}_{nat}(f, -1) = Pr.\{\mathcal{N}(0,1) \leq \overbrace{A - \sqrt{K \cdot A^2 + q(K)}}^{Z_{nat}(f,-1))}\}, \tag{3}$$

$$\mathcal{R}_{nat}(f, +1) = Pr.\{\mathcal{N}(0,1) \leq \underbrace{-K \cdot A + \sqrt{A^2 + q(K)}}_{Z_{nat}(f,+1))}\}, \tag{4}$$

*where* $A = \frac{2}{\sigma(K^2-1)}\sqrt{m\gamma^2 + d\eta^2}$ *and* $q(K) = \frac{2\log(K)}{K^2-1}$. *It has the robust error under attack* $||\delta|| \leq \epsilon_0$ :

$$\mathcal{R}_{rob}(f, -1) = Pr.\{\mathcal{N}(0,1) \leq \overbrace{Z_{nat}(f,-1) + \frac{m\gamma + d\eta}{\sqrt{m\gamma^2 + d\eta^2}}\frac{\epsilon_0}{\sigma}}^{Z_{rob}(f,-1))}\} \tag{5}$$

$$\mathcal{R}_{rob}(f, +1) = Pr.\{\mathcal{N}(0,1) \leq \underbrace{Z_{nat}(f,+1) + \frac{m\gamma + d\eta}{\sqrt{m\gamma^2 + d\eta^2}}\frac{\epsilon_0}{K\sigma}}_{Z_{rob}(f,+1))}\}. \tag{6}$$

Note that the term $A$ (consisting of feature dimension $d, m$ and center $\gamma, \eta$) represents how expressive the information from $\mathcal{D}$ that model $f$ can use for prediction. Thus, when the term $A$ is large, the model has close clean errors between the 2 classes, namely $\mathcal{R}_{nat}(f, -1) \approx \mathcal{R}_{nat}(f, +1)$. It is because the $q(K)$ term in their z-scores[2] can be ignored when $A$ is large. On the other hand for their robust errors, typically we assume the adversarial attack by $||\delta|| \leq \epsilon_0$ can bring in major threat to mislead natural model $f$, which will result both $Z_{rob}(f, -1)$ and $Z_{rob}(f, +1)$ to be large positive numbers, so that the robust errors of both classes $\mathcal{R}_{rob}(f, -1)$ and $\mathcal{R}_{rob}(f, +1)$ are also large.

## 3.3 Optimal Linear Model to Minimize Robust Error

During adversarial training, the desired linear classifier should minimize the total robust error, which minimizes the probability that there exists a perturbation $\delta$ constrained by budge $||\delta||_\infty \leq \epsilon$ that can let the model make mistake. Formally, we describe a linear classifier after adversarial training as:

$$f_{adv}^*(x) = \text{sign}(\langle w_{adv}^*, x \rangle + b_{adv}^*)$$
$$\text{where } w_{adv}^*, b_{adv}^* = \arg\min_{w,b} \Pr.(\exists \delta, ||\delta|| \leq \epsilon, \text{ s.t. } f_{adv}(x+\delta) \neq y). \tag{7}$$

---

[2]At here we use $Z_{nat}(f, -1)$ and $Z_{nat}(f, +1)$ to denote the z-score for the standard normal distribution corresponding to $\mathcal{R}_{nat}(f, -1)$ and $\mathcal{R}_{nat}(f, +1)$. Similarly, we use $Z_{rob}(f, -1)$ and $Z_{rob}(f, +1)$ to denote the z-score of the standard normal distribution of $\mathcal{R}_{rob}(f, -1)$ and $\mathcal{R}_{rob}(f, +1)$.

Similarly we use $f_{\text{adv}}$ to denote $f_{\text{adv}}^*$ for convenience. During adversarial training, the linear model will only preserve the weights on robust features and exclude all non-robust features as proved in Lemma 2 in the Appendix. We will also call the adversarially trained model in $\mathcal{D}$ a robust model, because its features cannot be easily manipulated by perturbations under $||\delta|| \leq \epsilon$. We show the robust model's clean and robust errors in Theorem 2 and the proof is provided in Appendix B.2.

**Theorem 2** *Optimal linear classifier which minimizes overall robust error in $\mathcal{D}$ will have class-conditional clean errors:*

$$\mathcal{R}_{nat}(f_{adv}, -1) = Pr.\{\mathcal{N}(0, 1) \leq B - \sqrt{K \cdot B^2 + q(K)} - \frac{\sqrt{m}}{\sigma}\epsilon\} \tag{8}$$

$$\mathcal{R}_{nat}(f_{adv}, +1) = Pr.\{\mathcal{N}(0, 1) \leq -K \cdot B + \sqrt{B^2 + q(K)} - \frac{\sqrt{m}}{K\sigma}\epsilon\} \tag{9}$$

*where $B = \frac{2}{\sigma(K^2-1)}\sqrt{m} \cdot (\gamma - \epsilon)$, $q(K) = \frac{2\log(K)}{K^2-1}$, and robust errors under attack $||\delta|| \leq \epsilon_0$:*

$$\mathcal{R}_{rob}(f_{adv}, -1) = Pr.\{\mathcal{N}(0, 1) \leq Z_{nat}(f_{adv}, -1) + \sqrt{m}\frac{\epsilon_0}{\sigma}\} \tag{10}$$

$$\mathcal{R}_{rob}(f_{adv}, +1) = Pr.\{\mathcal{N}(0, 1) \leq Z_{nat}(f_{adv}, +1) + \sqrt{m}\frac{\epsilon_0}{K\sigma}\}. \tag{11}$$

Recall that we assume the dimension of non-robust features is much higher than that of robust features ($d >> m$) and $\gamma$, $\eta$ and $\epsilon$ have similar scale: $\epsilon = \Theta(\gamma)$ and $\eta = \Theta(\gamma)$. Therefore, in Eq. (8) and Eq. (9), for model $f_{\text{adv}}$'s clean errors, the decisive term $B$ is at scale $\Theta((d/m)^{-\frac{1}{2}} \cdot A)$, where $A$ is the term in Eq.(3). In Corollary 1 (proved in Appendix B.3), we show that it is this relationship between $A$ and $B$ that will finally bring in the "unfair" issue to both model accuracy and robustness performance.

**Corollary 1** *Adversarially Trained Model on $\mathcal{D}$ will have larger clean error disparity between the 2 classes, compared to a Naturally Trained model.*

We calculate 2 classes' clean error difference: $(\mathcal{R}_{\text{nat}}(f, +1) - \mathcal{R}_{\text{nat}}(f, -1))$ for natural model $f$; and $(\mathcal{R}_{\text{nat}}(f_{\text{adv}}, +1) - \mathcal{R}_{\text{nat}}(f_{\text{adv}}, -1))$ for adversarially trained model $f_{\text{adv}}$. Since both terms are positive, we show their ratio (detailed proof in Appendix):

$$\Omega = \frac{\mathcal{R}_{\text{nat}}(f_{\text{adv}}, +1) - \mathcal{R}_{\text{nat}}(f_{\text{adv}}, -1)}{\mathcal{R}_{\text{nat}}(f, +1) - \mathcal{R}_{\text{nat}}(f, -1)} \geq \frac{(K^2-1)\log(K)\sigma^2}{2\Theta(\gamma^2)} \cdot \Theta(\frac{d}{m})^{\frac{1}{2}}. \tag{12}$$

In Eq. (12), we can tell once $d/m$ is large, the ratio $\Omega$ is also large (e.g. $> 1$) and the adversarial training is showed to enlarge the 2 classes' clean error disparity. The ratio $\Omega$ in Eq. (12) uncovers the main factor which may cause the "unfair" phenomenon in adversarially trained models: because the robust models make prediction in a feature space with dimension $m$ which is much lower than $d$ (dimension for natural models), their clean accuracy disparity between classes can be more sensitive to the classes' distributional difference $K$. In this way the robust model presents strong disparity of clean accuracy.

Furthermore, for the robust errors (i.e., Eqs. (10) and (42)) in adversariallly trained model, an adversarial attack under intensity $\epsilon_0$ will make the test error increase only by a small margin compared to the model's clean error in Eqs. (8) and (9). It is because the their z-scores difference is decided by $\sqrt{m}$ which is small and much lower than $d$. Because of the marginal difference between $R_{\text{nat}}(f_{\text{adv}}, -1)$ and $R_{\text{rob}}(f_{\text{adv}}, -1)$ (also for class "+1"), the model's fairness condition on robust errors will align to the clean errors (in Eq. (12)). Empirical results on real datasets also support this assumption (Figure 1). As a conclusion, adversarial training can bring in unfairness issues on both clean and robust performance.

## 4 FAIR ROBUST LEARNING (FRL)

Faced with the unfairness problem of adversarial training shown in Section 2 and 3, we desire to devise a Fair Robust Learning (FRL) strategy, in order to train robust models that have balanced accuracy and robustness performance for each class. Formally, we aim to train a classifier $f$ to have minimal overall robust error ($\mathcal{R}_{\text{rob}}(f)$); while stressing $f$ to satisfy a series of fairness constraints:

$$\begin{aligned} &\underset{f}{\text{minimize}} \quad \mathcal{R}_{\text{rob}}(f) \\ &\text{s.t.} \quad \mathcal{R}_{\text{nat}}(f, i) - \mathcal{R}_{\text{nat}}(f) \leq \tau_1 \text{ and } \mathcal{R}_{\text{rob}}(f, i) - \mathcal{R}_{\text{rob}}(f) \leq \tau_2 \text{ for each } i \in Y \end{aligned} \tag{13}$$

where $\tau_1$ and $\tau_2$ are small and positive values. The constraints in Eq. (13) restrict the model's error for each class $i$ (both clean error $\mathcal{R}_{\text{nat}}(f, i)$ and robust error $\mathcal{R}_{\text{rob}}(f, i)$) should not exceed the average level ($\mathcal{R}_{\text{nat}}(f)$ and $\mathcal{R}_{\text{rob}}(f)$) by a large margin. Therefore, the model will not have specific weak points under the risk of wrong prediction or adversarial attacking. Next, we will discuss the detailed components of our Fair Robust Learning (FRL) algorithm to solve the problem in Eq. (13).

## 4.1 TRADITIONAL MODEL DEBIASING METHOD: A REDUCTIONS APPROACH

In order to solve the fair robust training problem in Eq. (13), we follow the main pipeline from traditional machine learning debiasing works such as (Agarwal et al., 2018), which reduces the problem in Eq. (13) into a series of *Cost-sensitive* classification problems and continuously penalizes the terms which violate the fairness constraints. We begin by introducing Lagrange multipliers $\phi = (\phi_{\text{nat}}^i, \phi_{\text{rob}}^i)$ (non-negative) for each constraint in Eq. (13) and form the Lagrangian:

$$L(f, \phi) = \mathcal{R}_{\text{rob}}(f) + \sum_{i=1}^{Y} \phi_{\text{nat}}^i (\mathcal{R}_{\text{nat}}(f, i) - \mathcal{R}_{\text{nat}}(f) - \tau_1) + \sum_{i=1}^{Y} \phi_{\text{rob}}^i (\mathcal{R}_{\text{rob}}(f, i) - \mathcal{R}_{\text{rob}}(f) - \tau_2)$$

$$(14)$$

It equals to solving the max-min game between $f$ and $\phi$ as:

$$\max_{\phi_{\text{nat}}, \phi_{\text{rob}} \geq 0} \min_{f} \; L(f, \phi). \tag{15}$$

Typically, given a fixed $\phi$, if the current model $f$ violates some constraints in Eq. (13) (for example $\mathcal{R}_{\text{nat}}(f, i) - \mathcal{R}_{\text{nat}}(f) - \tau_1 > 0$), we first solve the outer-maximization problem in Eq. (15) by increasing its corresponding multiplier $\phi_{\text{nat}}^i$. As a result, we upweight the training weight (or cost) for the clean loss $\mathcal{R}_{\text{nat}}(f, i)$ of all samples in the class $i$. Then, the algorithm will solve the inner-minimization given new $\phi$ to optimize the model $f$, the error of $\mathcal{R}_{\text{nat}}(f, i)$ is therefore heavily penalized and the model will give more priority to the correct prediction for the class $i$. In this way the model gives more priority to mitigate violated terms in Eq. (14). During this process, the model $f$ and Lagrangian multiplier $\phi$ will be alternatively updated to achieve the equilibrium until we finally reach an optimal model that satisfies the fairness constraints. Based on this traditional debiasing strategy, next we will discuss the main difference of our task in the adversarial setting from this traditional approach.

## 4.2 DEBIAS CLEAN ERROR AND BOUNDARY ERROR SEPARATELY

One thing to note is that in the Eq (13), the robust error is always strongly related to clean errors (Zhang et al., 2019b; Tsipras et al., 2018) (see Eq. 16). Thus, during the debiasing process above, we could have twisted the influence on some class $i$'s clean and robust errors $\mathcal{R}_{\text{nat}}(f, i)$ and $\mathcal{R}_{\text{rob}}(f, i)$. It means that when we upweight the cost for $\mathcal{R}_{\text{rob}}(f, i)$ as introduced in Eq. (15), we also implicitly upweight the cost for $\mathcal{R}_{\text{nat}}(f, i)$. Thus, we will not get a precise update for $\phi$. To solve this issue, we can separate the robust error into the sum of *clean error* and *boundary error* inspired by (Zhang et al., 2019b) as:

$$\begin{aligned} \mathcal{R}_{\text{rob}}(f, i) &= \Pr.\{\exists \delta, \text{ s.t. } f(x + \delta) \neq y | y = i\} \\ &= \Pr.\{f(x) \neq y | y = i\} + \Pr.\{\exists \delta, f(x + \delta) \cdot f(x) \leq 0 | y = i\} \\ &= \mathcal{R}_{\text{nat}}(f, i) + \mathcal{R}_{\text{bdy}}(f, i) \end{aligned} \tag{16}$$

where $\mathcal{R}_{\text{bdy}}(f, i) = \Pr.\{\exists \delta, f(x + \delta) \cdot f(x) \leq 0 | y = i\}$ represents the probability that a sample from class $i$ lies close to the decision boundary and can be attacked. By separating the clean error and boundary error during adversarial training, we are able to independently debias the unfairness from both clean error and boundary error. Formally, we have the training objective as:

$$\begin{aligned} \underset{f}{\text{minimize}} \quad & \mathcal{R}_{\text{nat}}(f) + \mathcal{R}_{\text{bdy}}(f) \\ \text{s.t.} \quad & \mathcal{R}_{\text{nat}}(f, i) - \mathcal{R}_{\text{nat}}(f) \leq \tau_1 \text{ and } \mathcal{R}_{\text{bdy}}(f, i) - \mathcal{R}_{\text{bdy}}(f) \leq \tau_2 \text{ for each } i \in Y \end{aligned} \tag{17}$$

We introduce Lagrangian multipliers $\phi = (\phi_{\text{nat}}^i, \phi_{\text{bdy}}^i)$ and solve the max-min game for Eq. (17) similar to Eq. (15). Note that if the constraints in Eq. (17) are satisfied, the fairness quality for robust error ($\mathcal{R}_{\text{rob}}(f, i) = \mathcal{R}_{\text{nat}}(f, i) + \mathcal{R}_{\text{bdy}}(f, i)$) of each class can also be guaranteed by $\tau_1 + \tau_2$. In practice, we will use surrogate loss functions (such as cross entropy) $\mathcal{L}(f(x), y)$ and $\max_{||\delta|| \leq \epsilon} \mathcal{L}(f(x), f(x'))$ to optimize the clean and boundary errors as suggested by (Zhang et al., 2019b).

### 4.3 Cost-Sensitive Classification for Clean Errors vs Boundary Errors

During the debiasing training process in Eq. (17), if one class $i$'s clean error violates the fairness inequality: $\mathcal{R}_{\text{nat}}(f,i) - \mathcal{R}_{\text{nat}}(f) - \tau_1 > 0$, upweighting the cost for $\mathcal{R}_{\text{nat}}(f,i)$ can help penalize large $\mathcal{R}_{\text{nat}}(f,i)$ and mitigate the unfairness issue as suggested by (Agarwal et al., 2018). Note that we refer to this strategy as "Reweight". However, if the boundary error for class $i$: $\mathcal{R}_{\text{bdy}}(f,i) - \mathcal{R}_{\text{bdy}}(f) - \tau_2 > 0$, we claim that only upweighting its cost (or Reweight) could not succeed to fulfill the cost-sensitive classification goal in adversarial setting. Our empirical studies in Section 5 show that upweighting the boundary error for some class $i$ cannot effectively reduce model's boundary error specifically for class $i$ but can bring in side-effects to degrade class $i$'s clean accuracy. It is evident from (Ding et al., 2018) that increasing the margin $\epsilon$ during adversarial training can effectively improve model's robustness against attacks under current intensity $\epsilon$. Therefore, we hypothesize that enlarging the adversarial margin $\epsilon$ when generating adversarial examples during training specifically for the class $i$ can improve this class's robustness and reduce the large boundary error $\mathcal{R}_{\text{bdy}}(f,i)$. In this work, we refer to this strategy as "Re-margin". Empirical study in Table 2 and Figure 3 validates its effectiveness.

We present the main components and process of Fair Robust Learning (FRL) in Algorithm 1. Note that $\text{BEST}(f, \phi, \epsilon)$ denotes the adversarial training process under adjusted hyper-parameter $\phi, \epsilon$ mentioned in Eq. (14) and we test the performance under the validation set which is denoted as $\text{EVAL}(f, \cdot)$. In Algorithm 1, in each iteration we first test our initialized or pretrained model $f$ on the validation set to check whether it violates the unfairness constraints (step 5). Then we update Lagrangian multiplier $\phi_{\text{nat}}$ to reweight the clean loss for each class. We propose three strategies to update hyper-parameter to balance the boundary loss including Reweight (option 1), Remargin (option 2) and Reweight+Regmargin (option 3). We follow one of the options (step 7) for boundary loss. Finally we adversarially train the model under the updated setting by $\phi, \epsilon$.

---

**Algorithm 1** The Fair Robust Learning (FRL) Algorithm

---

1: **Input:** Fairness constraints specified by $\tau_1 > 0$ and $\tau_2 > 0$, test time attacking radius $\epsilon_0$ and hyper-param update rate $\alpha_1, \alpha_2$

2: **Output:** Fairly robust neural network $f$

3: Randomly initialize network $f$ or initialize network with pre-trained configuration

    Set $\phi_{\text{nat}}^i = 0$, $\phi_{\text{bdy}}^i = 0$ and $\phi = (\phi_{\text{nat}}, \phi_{\text{bdy}})$, adv. training radius $\epsilon_i = \epsilon_0$ for each $i \in \mathcal{Y}$

4: **repeat**

5:     $\mathcal{R}_{\text{nat}}(i, f), \mathcal{R}_{\text{bdy}}(i, f) = \text{EVAL}(f, \epsilon_0)$                $\triangleright$ Evaluate $f$ for each class

6:     $\phi_{\text{nat}}^i = \phi_{\text{nat}}^i + \alpha_1 \cdot (\mathcal{R}_{\text{nat}}(i, f) - \tau_1)$          $\triangleright$ Update multiplier $\phi_{\text{nat}}$

7:     $\phi_{\text{bdy}}^i = \phi_{\text{bdy}}^i + \alpha_2 \cdot (\mathcal{R}_{\text{bdy}}(i, f) - \tau_2)$      $\triangleright$ Option 1. Update multiplier $\phi_{\text{bdy}}$

7:     **or** $\phi_{\text{bdy}}^i = \phi_{\text{bdy}}^i$; $\epsilon_i = \epsilon_i \cdot \exp(\alpha_2 \cdot (\mathcal{R}_{\text{bdy}}(i, f) - \tau_2))$       $\triangleright$ Option 2. Remargin

7:     **or** $\phi_{\text{bdy}}^i = \phi_{\text{bdy}}^i + \alpha_2 \cdot (\mathcal{R}_{\text{bdy}}(i, f) - \tau_2)$ ;

        $\epsilon_i = \epsilon_i \cdot \exp(\alpha_2 (\mathcal{R}_{\text{bdy}}(i, f) - \tau_2))$      $\triangleright$ Option 3. Reweight + Remargin

8:     $f \leftarrow \text{BEST}(f, \phi, \epsilon)$    $\triangleright$ Adv. training under hyper-param $\phi, \epsilon$ from current $f$

9: **until** Model $f$ satisfies all constraints

---

## 5 Experiment

In this section, we will present the experimental results to validate the effectiveness of the proposed framework (FRL) for building fairly robust DNN models. We implement and compare our proposed three strategies (i.e., Reweight, Remargin and Reweight+Remargin) on real-world data and discuss their possible different consequences. We also discuss the main difference of the manner of our proposed three potential debiasing strategies.

**Experimental Setup & Baselines** We conduct our experiments on CIFAR10 (Krizhevsky et al., 2009). For CIFAR10, we present our main results under the model architecture PreAct Residual Network (He et al., 2016). As comparison sets to show our method can improve fairness, we also present the original performance from two popular adversarial training algorithms (Madry et al., 2017; Zhang et al., 2019b). Meanwhile we add a baseline debiasing method which are inherited from (Agarwal et al., 2018) (we directly apply it to reweight the cost of adversarial examples during

adversarial training) as a representative to show traditional debiasing methods might not be easily applied to solve unfairness issues in adversarial setting. For CIFAR10 dataset, we mainly test our method's performance on a ResNet18 (He & Garcia, 2009)) model, and we apply the PGD attack (Madry et al., 2017) algorithm to generate adversarial examples under $l_\infty$-norm under $8/255$. For each of the debiasing algorithm, we set the fairness constraints $\tau_1$ and $\tau_2$ be $5\%$ and $7\%$ respectively, for clean and boundary errors. Please refer to this link[3] for our empirical implementations.

**Debiasing Performance (CIFAR10)** We first check whether our proposed FRL framework can help resolve the unfairness issue in adversarial training. Refer to our goal in Eq. (13) to achieve the fairness constraints to get both balanced clean and robustness performance. We report the trained model's average *clean error* rate, *boundary error* rate and *robust error* rate (defined in Eq.(16)), as well as the worst intra-class clean, boundary and robust error rates (Table 1). Thus, for an optimal equally robust model, we hope each of these worst intra-class errors is not too large compared to the average errors. Meanwhile, it is also necessary that one debiasing strategy should not have too much sacrifice on the model's overall clean and robustness performance.

Table 1: Average & worst-class clean error, boundary error and robust error for various algorithms.

| | Avg. Clean | Worst Clean | Avg. Bdy. | Worst Bdy. | Avg. Rob. | Worst Rob. |
|---|---|---|---|---|---|---|
| **PGD Adv. Training** | 17.3 | 40.9 | 39.4 | 54.4 | 56.9 | 82.6 |
| **TRADES**($\beta = 1$) | **14.4** | 27.9 | 43.6 | 62.6 | 58.0 | 83.6 |
| **TRADES**($\beta = 2$) | 16.9 | 34.9 | **39.1** | 56.6 | **55.5** | 82.2 |
| **Baseline Reweight** | 19.2 | 28.3 | 39.2 | 53.7 | 58.2 | 80.1 |
| **FRL(Reweight)** | 17.0 | **22.5** | 41.6 | 51.2 | 58.6 | 73.3 |
| **FRL(Remargin)** | 16.9 | 24.9 | 41.6 | 50.6 | 58.5 | 76.3 |
| **FRL(Reweight+Remargin)** | 18.4 | 24.7 | 40.3 | **47.4** | 58.7 | **70.2** |

In Table 1 for CIFAR10 dataset, we present the performance of all three versions of our proposed FRL framework. Compared to the baselines, all the FRL algorithms reduce the worst intra-class clean, boundary and robust error under different degrees. FRL (Reweight) can get the best debiasing performance to achieve minimal "worst-class" clean error, but it cannot debias boundary loss well. The method (Reweight + Remargin) can be the most effective way to debias boundary error disparity and robust error disparity.Our added baseline method (Baseline Reweight) (Agarwal et al., 2018) only have minor help for clean performance fairness but cannot debias boundary error or robust error. For each debiasing method of FRL, compared to vanilla PGD and TRADES, the total average performance is only degraded by a slight margin ($1 \sim 2\%$ for clean error and $1 \sim 3\%$ for robust error), thus the debiasing method will not sacrifice too much total performance.

**Debiasing Performance (GTSRB)** We also apply our FRL debiasing method to GTSRB (Stallkamp et al., 2011), which consists of images from 43 various traffic signs. Since this dataset is highly unbalanced (some classes only have around 200 training images), we only keep 17 classes when performing adversarial training. For this dataset, we apply a 4-layer CNN classifier and test the robustness using PGD attack under $l_\infty$-norm by $12/255$. In Table 2, we list the average and worst-class clean, robust and boundary errors for different training algorithms. The results show that the FRL method (Reweight, Reweight+Remargin) can effectively help to improve the model's worst-class clean and robust performance. Furthermore, since the original dataset is imbalanced, the debiasing method brings in side-effect to improve the model's overall performance, with lower average clean error and robust error.

Table 2: Average & worst-class clean, boundary and robust error for various algorithms in GTSRB.

| | Avg. Clean | Worst Clean | Avg. Bdy. | Worst Bdy. | Avg. Rob. | Worst Rob. |
|---|---|---|---|---|---|---|
| **PGD Adv. Training** | 1.2 | 9.1 | 18.3 | 42.3 | 20.6 | 49.5 |
| **TRADES**($\beta = 1$) | 1.6 | 7.3 | 21.5 | 50.4 | 23.1 | 59.7 |
| **TRADES**($\beta = 2$) | 2.1 | 10.3 | 17.5 | 40.3 | 19.6 | 46.3 |
| **Baseline Reweight** | 0.8 | 6.4 | 19.4 | 42.4 | 20.1 | 44.0 |
| **FRL(Reweight)** | **0.7** | **3.5** | 18.3 | 39.4 | 19.0 | 40.0 |
| **FRL(Remargin)** | 2.4 | 10.7 | 18.0 | 44.3 | 18.4 | 44.3 |
| **FRL(Reweight+Remargin)** | 0.8 | 4.0 | **16.6** | **34.2** | **17.5** | **38.2** |

---

[3]https://drive.google.com/open?id=1IjE11VpV5CGXx633Nef3McgVOoRGMK9x

**Compare Different Debiasing Strategies.** In Figure 3, we take a closer look at the behavior of different proposed debiasing strategies mentioned in Section 4.3 and test whether they can succeed in solving the constrained training problem in Eq.(17). We present the model's maximum violation (e.g. $v(i) =$

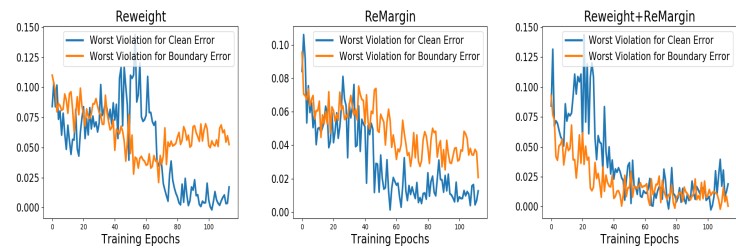

Figure 3: Debiasing Manner for 3 FRL Options in CIFAR10.

$\mathcal{R}_{\text{nat}}(f, i) - \mathcal{R}_{\text{nat}}(f, i) - \tau_1$) among all classes for each training epoch that hot-started from a pretrained (adversarially) ResNet model. If $v(i) \leq 0$ for each class in $\mathcal{Y}$, each fairness constraint is satisfied. From the figure we can tell that FRL (Reweight) method cannot adequately balance the boundary error, and it always presents a trade-off relation with the clean error constraints. Introducing the Remargin method will facilitate to achieve fairness for boundary errors. More details (e.g. average and worst clean / robustness in training) are in Fig. 5 in Appendix A.2.

## 6   RELATED WORKS

**Adversarial Attacks and Adversarial Training.** The existence of adversarial attacks (Goodfellow et al., 2014; Szegedy et al., 2013; Carlini & Wagner, 2017) causes huge concerns when people adopt machine learning models in various application domains (Xu et al., 2019; Jin et al., 2020). As countermeasures against adversarial examples, adversarial training (robust optimization) algorithms (Goodfellow et al., 2014; Madry et al., 2017; Zhang et al., 2019b; Shafahi et al., 2019; Zhang et al., 2019a) are formulated as a min-max problem that directly minimize the model's risk on the adversarial samples such that the machine learning model is robust under the adversarial attacks. They are shown to be one of the most reliable strategies to improve model safety. Another mainstream of defense methods are certified defense, which aims to provide provably robust DNNs under $l_p$ norm bound (Wong & Kolter, 2018; Cohen et al., 2019) and guarantee the robustness. In this work, we focus on studying the potential risk of the defense algorithms from a new scope of the fairness concerns.

**Fairness in Machine Learning & Imbalanced Dataset.** Fairness issues recently draw much attention from the community of machine learning. These issues can generally divided into two categorizations: (1) prediction outcome disparity: the models tend to have some unreasonable preference of prediction for some specific demographic groups (Zafar et al., 2017); and (2) prediction quality disparity: the models tend to have much lower performance on some specific groups than others (Buolamwini & Gebru, 2018). Please refer to the works (Barocas et al.; Mehrabi et al., 2019) for a more comprehensive and detailed summary of fairness study in machine learning. The reasons that cause these discrimination problems might come from data distribution or the learning algorithms. Unlike existing works, this work is the first to study the unfairness issue in the adversarial setting. We argue that robustly trained models are likely to have different accuracy and robustness quality among different classes, and it may be introduced by both data distribution and the adversarial training algorithm. We also mention imbalanced data learning problem (He & Garcia, 2009; Lin et al., 2017) as one related topic of our work. Since in our work, (e.g. Figure 1), we show that the prediction performance differences are indeed existing between different classes. This phenomenon is also well-studied in imbalanced data problems or long-tail distribution learning problems (Wang et al., 2017) where some classes have much fewer training samples than others. However, in our case, we show that this unfairness problem can also happen in the balanced data, so it desires new scopes and methods for a further study.

## 7   CONCLUSION

In this work we first theoretically and empirically uncover one side-effect of adversarial training algorithms: it can cause serious disparity for both clean accuracy and adversarial robustness between different classes of the data. As a first attempt to resolve unfairness issues from adversarial training, we propose the Fair Robust Learning (FRL) framework to mitigate this issue.

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

## A   APPENDIX.

### A.1   OVERALL PERFORMANCE ON MORE MODELS

Table 3:   Adversarial training algorithms on CIFAR10 dataset (ResNet18 (above) and ReNet34(below)). We report the average clean accuracy and adversarial accuracy (under PGD attack by $8/255$), as well as the worst / best clean accuracy and adv. accuracy among all classes.

|  | Avg. Clean. | Avg. Adv. | Worst Clean | Best Clean | Worst Adv. | Best Adv. |
|---|---|---|---|---|---|---|
| **Natural Training** | 93.1 | 0.0 | 87.5 | 97.7 | 0.0 | 0.0 |
| **PGD Adv. Training** | 82.7 | 43.1 | 59.1 | 93.6 | 17.4 | 64.9 |
| **TRADES** | 83.1 | 44.0 | 65.1 | 93.4 | 17.8 | 67.0 |
| **Natural Training** | 95.1 | 0.0 | 88.1 | 98.2 | 0.0 | 0.0 |
| **PGD Adv. Training** | 86.6 | 46.3 | 72.3 | 96.4 | 19.8 | 72.2 |
| **TRADES** | 85.5 | 56.3 | 67.0 | 95.2 | 27.5 | 79.5 |

### A.2   THE PHENOMENON OF UNFAIR ROBUSTNESS ON GTSRB

**GTSRB** We also show the similar unfairness phenomenon in German Traffic Sign Recognition Benchmark (GTRSB) (Stallkamp et al., 2011), which consist of 43 classes of images from different traffic signs, with image sizes $32 \times 32 \times 3$. In this dataset we also both naturally and adversarially train a 3-Layer CNN classifier. We list the model's performance and sort the classes in the order of decreasing clean accuracy and adv. accuracy. From the Figure 4, we also see the natural training has similar clean accuracy between different classes, but adversarial training will enlarge their gap and

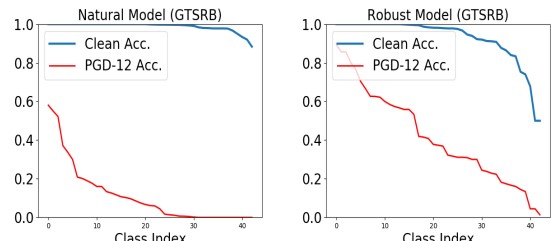

Figure 4: Unfairness in GTSRB

result the clean performance has a heavier tail property. In this dataset, both natural model and robust model have clear distinguished adversairal accuracy (robustness). However, adversarial training even hardly provide any robustness improvement for some classes.

|  | Avg. Clean. | Avg. Adv. | Worst Clean | Best Clean | Worst Adv. | Best Adv. |
|---|---|---|---|---|---|---|
| **Natural Training** | 99.5 | 18.9 | 91.7 | 100.0 | 0.0 | 72.5 |
| **PGD Adv. Training** | 94.5 | 44.4 | 50.0 | 100.0 | 1.3 | 90.0 |
| **TRADES** | 91.2 | 47.2 | 35.3 | 100.0 | 3.3 | 92.0 |

Table 4: Adversarial training algorithms on GTSRB dataset (on a 3-layer CNN model). We report the average clean accuracy and adversarial accuracy (under PGD attack by $12/255$), as well as the worst / best clean accuracy and adv. accuracy among all classes.

## B   PROOF OF THEOREMS

### B.1   PROOF OF THEOREM 1

Before proving Theorem 1, we first establish the optimal linear classifier in natural training through Lemma 1, which facilitates us to prove the clean error and robust error after natural training.

**Lemma 1 (Optimal linear classifier in natural training)** *For the data following the distribution in Eq. (1), the naturally trained linear classifier $f(x) = sign(w^T x + b)$ has the optimal weight that satisfy: $w_1 = w_2 = \cdots = w_m$ and $w_{m+1} = w_{m+2} = \cdots = w_{m+b}$. Moreover, their ratio satisfy $w_1 : w_{m+1} = \gamma : \eta$ and $b : w_1 = w_0 : 1$ where*

$$w_0 = A^2 \frac{K^2+1}{K^2-1} - A^2 K \sqrt{\frac{4}{(K^2-1)^2} + \frac{2\sigma^2 \log K}{A^2(K^2-1)}}.$$

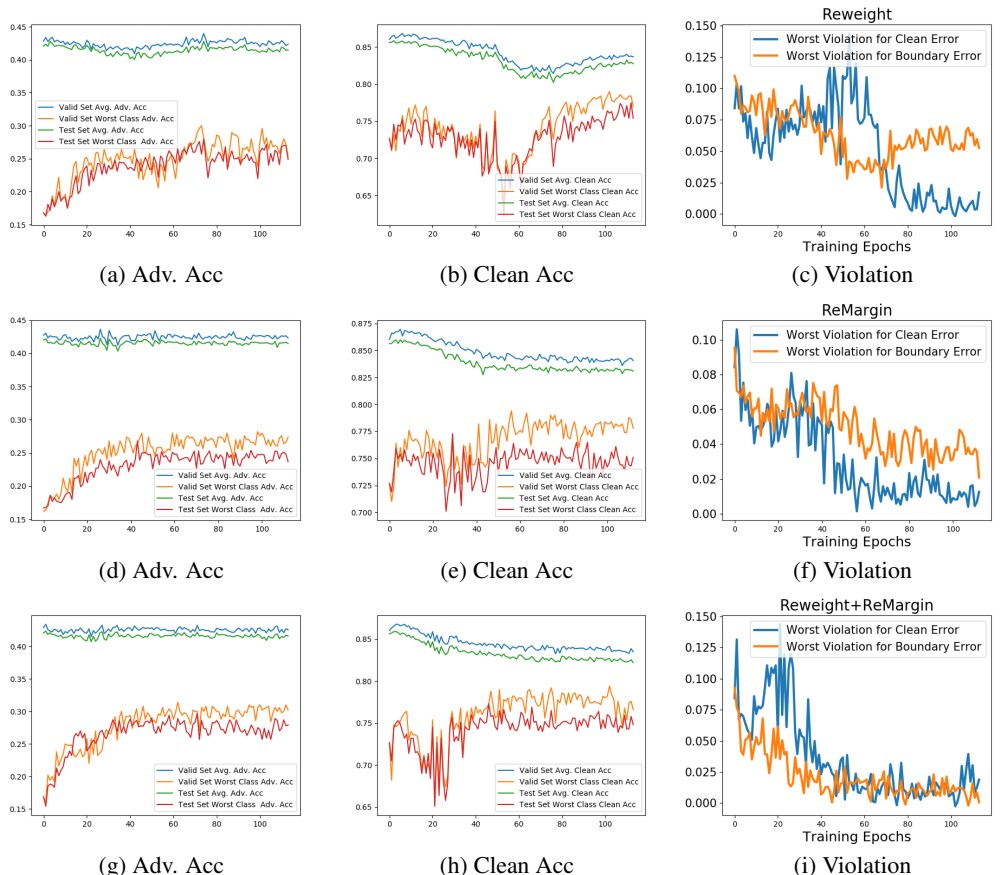

Figure 5: Debiasing training process

**Proof 1 (Proof of Lemma 1)** *We first prove $w_1 = w_2 = \cdots = w_m$ and $w_{m+1} = w_{m+2} = \cdots = w_{m+b}$ by contradiction. We define $G_1 = \{1, 2, \ldots, m\}$ and $G_2 = \{m+1, m+2, \ldots, m+d\}$. We make the following assumption: for the optimal $w$ and $b$, we assume there exist $w_i < w_j$ for $i \neq j$ and $i, j \in G_1$. Then we obtain the following clean error for two classes with this classifier $w$*

$$R(f, -1) = Pr\{\sum_{k \neq i, k \neq j} w_k \mathcal{N}_k + b + w_i \mathcal{N}(-\gamma, \sigma_{-1}^2) + w_j \mathcal{N}(-\gamma, \sigma_{-1}^2) > 0\} \tag{18}$$

$$R(f, +1) = Pr\{\sum_{k \neq i, k \neq j} w_k \mathcal{N}_k + b + w_i \mathcal{N}(+\gamma, \sigma_{+1}^2) + w_j \mathcal{N}(+\gamma, \sigma_{+1}^2) < 0\} \tag{19}$$

*If we use $w_j$ to replace $w_i$, we obtain new errors*

$$\overline{R(f, -1)} = Pr\{\sum_{k \neq i, k \neq j} w_k \mathcal{N}_k + b + w_j \mathcal{N}(-\gamma, \sigma_{-1}^2) + w_j \mathcal{N}(-\gamma, \sigma_{-1}^2) > 0\} \tag{20}$$

$$\overline{R(f, +1)} = Pr\{\sum_{k \neq i, k \neq j} w_k \mathcal{N}_k + b + w_j \mathcal{N}(+\gamma, \sigma_{+1}^2) + w_j \mathcal{N}(+\gamma, \sigma_{+1}^2) < 0\}. \tag{21}$$

*It implies $\overline{R(f, -1)} + \overline{R(f, +1)} < R(f, -1) + R(f, +1)$.*

*Therefore, it contradicts with the assumption we make and we conclude for an optimal linear classifier in natural training, it must satisfies $w_1 = w_2 = \cdots = w_m$. The same argument applies to $i, j \in G_2$ and we conclude $w_{m+1} = w_{m+2} = \cdots = w_{m+d}$ similarly.*

Next, we calculate the ratio between $w_1, w_{m+1}$.

$$R(f) = Pr\{f(x) \neq y\} \tag{22}$$
$$= Pr\{f(x) = 1|y = -1\} \cdot Pr\{y = -1\} + Pr\{f(x) = -1|y = 1\} \cdot Pr\{y = +1\}$$
$$\propto Pr\{w^T x + b > 0|y = -1\} + Pr\{w^T x + b < 0|y = -1\}$$
$$= Pr\{\sum_{i \in G_1} w_i \mathcal{N}(-\gamma, \sigma_{-1}^2) + \sum_{j \in G_2} w_j \mathcal{N}(-\eta, \sigma_{-1}^2) + b > 0|y = -1\}$$
$$+ Pr\{\sum_{i \in G_1} w_i \mathcal{N}(+\gamma, \sigma_{+1}^2) + \sum_{j \in G_2} w_j \mathcal{N}(+\eta, \sigma_{+1}^2) + b < 0|y = +1\}$$
$$= Pr\{\mathcal{N}(0, 1) < \underbrace{\frac{b - mw_1\gamma - dw_{m+1}\eta}{\sqrt{mw_1^2 + dw_{m+1}^2}\sigma}}_{Z(-1)}\} + Pr\{\mathcal{N}(0, 1) < \underbrace{\frac{-b - mw_1\gamma - dw_{m+1}\eta}{\sqrt{mw_1^2 + dw_{m+1}^2}\sigma K}}_{Z(+1)}.\}$$

$$\tag{23}$$

We derive the optimal $w_1$ and $w_{m+1}$ by taking $\frac{\partial R(f)}{\partial w_1} = 0$ and $\frac{\partial R(f)}{\partial w_{m+1}} = 0$:

$$\frac{\partial R(f)}{\partial w_1} = \frac{1}{\sqrt{2\pi}} \exp\left(-\frac{1}{2}\left(Z(-1)\right)^2\right) \cdot \frac{\partial Z(-1)}{\partial w_1} + \frac{1}{\sqrt{2\pi}} \exp\left(-\frac{1}{2}\left(Z(+1)\right)^2\right) \cdot \frac{\partial Z(+1)}{\partial w_1} = 0$$

$$\frac{\partial R(f)}{\partial w_{m+1}} = \frac{1}{\sqrt{2\pi}} \exp\left(-\frac{1}{2}\left(Z(-1)\right)^2\right) \cdot \frac{\partial Z(-1)}{\partial w_{m+1}} + \frac{1}{\sqrt{2\pi}} \exp\left(-\frac{1}{2}\left(Z(+1)\right)^2\right) \cdot \frac{\partial Z(+1)}{\partial w_{m+1}} = 0$$

These imply

$$\frac{\partial Z(-1)}{\partial w_1} \Big/ \frac{\partial Z(+1)}{\partial w_1} = \frac{\partial Z(-1)}{\partial w_{m+1}} \Big/ \frac{\partial Z(+1)}{\partial w_{m+1}}$$

and therefore

$$\frac{-\gamma dw_{m+1}^2 - bw_1 + d\eta w_1 w_{m+1}}{-\gamma dw_{m+1}^2 + bw_1 + d\eta w_1 w_{m+1}} = \frac{-\eta mw_1^2 - bw_2 + m\gamma w_1 w_{m+1}}{-\eta mw_1^2 + bw_{m+1} + m\gamma w_1 w_{m+1}}. \tag{24}$$

Simplifying it gives $w_1 : w_{m+1} = \eta : \gamma$.

Then, we calculate the ratio between $w_1$ and $b$. Based on the relation between $w_1$ and $w_{m+1}$, we let $w_1 = \gamma w$ and $w_{m+1} = \eta w$ for some constant w. Substitute $w_1$ and $w_{m+1}$ into Eq. (23), we have

$$R(f, -1) = Pr\{\mathcal{N}(0, 1) < \frac{b}{\sigma w\sqrt{w\gamma^2 + d\eta^2}} - \frac{m\gamma^2 w + d\eta^2 w}{\sigma w\sqrt{m\gamma^2 + d\eta^2}}\} \tag{25}$$

$$= Pr\{\mathcal{N}(0, 1) < \frac{b}{\sigma w\sqrt{w\gamma^2 + d\eta^2}} - \frac{\sqrt{m\gamma^2 + d\eta^2}}{\sigma}\}, \tag{26}$$

$$R(f, +1) = Pr\{\mathcal{N}(0, 1) < \frac{-b}{\sigma wK\sqrt{w\gamma^2 + d\eta^2}} + \frac{\sqrt{m\gamma^2 + d\eta^2}}{K\sigma}\}. \tag{27}$$

For simplicity, we denote $A = \sqrt{m\gamma^2 + d\eta^2}$ and assume $w_o = \frac{b}{w}$. We will find optimal b by letting $\frac{\partial[R(f, -1) + R(f, +1)]}{\partial w_0} = 0$. In detail, it is

$$\frac{1}{A\sigma\sqrt{2\pi}} \exp(-\frac{1}{2}(\frac{w_0}{A\sigma} - \frac{A}{\sigma})^2) - \frac{1}{KA\sigma\sqrt{2\pi}} \exp(-\frac{1}{2}(\frac{w_0}{KA\sigma} - \frac{A}{K\sigma})^2) = 0 \tag{28}$$

which gives $\frac{1}{2}(\frac{w_0}{KA\sigma} + \frac{A}{K\sigma})^2 + \frac{1}{2}(\frac{w_o}{A\sigma} - \frac{A}{\sigma})^2 = 2\log K$ and therefore we obtain

$$w_0 = A^2 \frac{K^2 + 1}{K^2 - 1} - A^2 K\sqrt{\frac{4}{(K^2 - 1)^2} + \frac{2\sigma^2 \log K}{A^2(K^2 - 1)}}.$$

**Proof 2 (Proof of Theorem 1)** *Substitute the optimal linear model $\{w, b\}$ in Eq. (23), we have clean errors*

$$\mathcal{R}_{nat}(f, -1) = Pr.\{\mathcal{N}(0, 1) \leq Z_{nat}(f, -1)\}, \tag{29}$$

$$\mathcal{R}_{nat}(f, +1) = Pr.\{\mathcal{N}(0, 1) \leq Z_{nat}(f, +1)\}, \tag{30}$$

*where*

$$Z_{nat}(f, -1)\} = \frac{2A}{(K^2 - 1)\sigma} - K\sqrt{\frac{4A^2}{(K^2 - 1)^2} + \frac{2 \log K}{(K^2 - 1)}}, \tag{31}$$

$$Z_{nat}(f, +1)\} = \frac{-2KA}{(K^2 - 1)\sigma} + \sqrt{\frac{4A^2}{(K^2 - 1)^2 \sigma^2} + \frac{2 \log K}{(K^2 - 1)}}. \tag{32}$$

*Substitute the optimal linear model $\{w, b\}$ into robust error in Eq. (35) and (36), we have robust errors:*

$$\mathcal{R}_{rob}(f, -1) = Pr.\{\mathcal{N}(0, 1) \leq Z_{nat}(f, -1) + \frac{m\gamma + d\eta}{\sqrt{m\gamma^2 + d\eta^2}} \frac{\epsilon_0}{\sigma}\}, \tag{33}$$

$$\mathcal{R}_{rob}(f, +1) = Pr.\{\mathcal{N}(0, 1) \leq Z_{nat}(f, +1) + \frac{m\gamma + d\eta}{\sqrt{m\gamma^2 + d\eta^2}} \frac{\epsilon_0}{K\sigma}\}. \tag{34}$$

### B.2 PROOF OF THEOREM 2

Before proving Theorem 2, we first establish the optimal linear classifier in adversarial training through Lemma 2, which facilitates us to prove the clean error and robust error after adversarial training.

**Lemma 2 (Optimal linear model in adversarial training)** *For the data distribution assumed in Eq. (1), the adversarially trained linear classifier $f(x) = sign(w^T x + b)$ has the optimal weight that satisfy: $w_1 = w_2 = \cdots = w_m$, $w_{m+1} = w_{m+2} = \cdots = w_{m+b} = 0$ and $b : w_1 = w_0 : 1$ where*

$$w_0 = m(\gamma - \epsilon)^2 \frac{K^2 + 1}{K^2 - 1} - m(\gamma - \epsilon)^2 K \sqrt{\frac{4}{(K^2 - 1)^2} + \frac{2\sigma^2 \log K}{m(\gamma - \epsilon)^2 (K^2 - 1)}}.$$

**Proof 3 (Proof of Lemma 2)**

$$\begin{aligned}
R_{rob}(f, -1) &= Pr.\{\exists \|\delta\| \leq \epsilon, f(x + \delta) > 0 \mid y = -1\} \\
&= Pr.\{\max_{\|\delta\| \leq \epsilon} f(x + \delta) > 0 \mid y = -1\} \\
&= Pr.\{\max_{\|\delta\| \leq \epsilon} \sum_{i=1}^{m} w_i(\mathcal{N}(-\gamma, \sigma_{-1}^2) + \delta_i) + \sum_{i=m+1}^{m+d} w_i(\mathcal{N}(-\eta, \sigma_{-1}^2) + \delta_i) + b > 0\}.
\end{aligned} \tag{35}$$

*Similarly, we have*

$$\begin{aligned}
R_{rob}(f, +1) &= Pr.\{\exists \|\delta\| \leq \epsilon, f(x + \delta) < 0 | y = +1\} \\
&= Pr.\{\max_{\|\delta\| \leq \epsilon} f(x + \delta) > |y = +1\} \\
&= Pr.\{\max_{\|\delta\| \leq \epsilon} \left\{ \sum_{i=1}^{m} w_i(\mathcal{N}(\gamma, \sigma_{+1}^2) + \delta_i) + \sum_{i=m+1}^{m+d} w_i(\mathcal{N}(+\eta, \sigma_{+1}^2) + \delta_i) \right\} + b < 0\}.
\end{aligned} \tag{36}$$

To prove the theorem by contradiction, we first assume that for the optimal $w$, there exist some $w_i > 0$ where $i \in G_2 = \{m+1, m+2, \ldots, m+d\}$. Then

$$R_{rob}(f, -1) = Pr.\{\underbrace{\sum_{j \neq i} \max_{|\delta_j| \leq \epsilon} w_j(\mathcal{N}(\theta_j, \sigma_{-1}^2) + \delta_j) + b}_{\mathbb{A}} + \underbrace{\max_{|\delta_i| \leq \epsilon} w_i(\mathcal{N}(-\eta, \sigma_{-1}^2) + \delta_i)}_{\mathbb{B}} > 0\}. \quad (37)$$

Because $w_i > 0$, $\mathbb{B}$ is maximized when $\delta = \epsilon$. We obtain

$$R_{rob}(f, -1) = Pr.\{\mathbb{A} + w_i N(-\eta + \epsilon, \sigma_{-1}^2) > 0\} \geq Pr.\{\mathbb{A} > 0\} \quad (38)$$

So $w_i = 0$ gives a better robust error. We can also assume $w_i < 0$ and use similar contradiction to prove $w_i = 0$. Similar argument holds for $R_{rob}(f, +1)$. Therefore, we arrive at the conclusion $w_i = 0$ for all $m+1 \leq i \leq m+d$. The calculation of $w_0$ is similar to the proof of Lemma 1 and we omit the proof here.

**Proof 4 (Proof of Theorem 2)** *Substitute the optimal linear model in Lemma 2 into Eq. (23), we obtain the clean error on the adversarially trained model:*

$$R_{nat}(f, -1) = Pr.\{N(0,1) < \underbrace{\frac{2\sqrt{m}(\gamma - \epsilon)}{(K^2 - 1)\sigma} - K\sqrt{\frac{4m(\gamma - \epsilon)^2}{(K^2 - 1)^2 \sigma^2} + \frac{2 \log K}{K^2 - 1}}}_{Z_{nat}(f_{adv}, -1)}\}, \quad (39)$$

$$R_{nat}(f, +1) = Pr.\{N(0,1) < \underbrace{-\frac{2\sqrt{m}(\gamma - \epsilon)}{(K^2 - 1)\sigma} + K\sqrt{\frac{4m(\gamma - \epsilon)^2}{(K^2 - 1)^2 \sigma^2} + \frac{2 \log K}{K^2 - 1}}}_{Z_{nat}(f_{adv}, +1)}\}. \quad (40)$$

*Substitute the optimal linear model in Lemma 2 into Eq. (35) and (36), we obtain the robust error on the adversarially trained model:*

$$\mathcal{R}_{rob}(f_{adv}, -1) = Pr.\{\mathcal{N}(0,1) \leq Z_{nat}(f_{adv}, -1) + \sqrt{m}\frac{\epsilon_0}{\sigma}\}, \quad (41)$$

$$\mathcal{R}_{rob}(f_{adv}, +1) = Pr.\{\mathcal{N}(0,1) \leq Z_{nat}(f_{adv}, +1) + \sqrt{m}\frac{\epsilon_0}{K\sigma}\}. \quad (42)$$

### B.3 PROOF OF COROLLARY 1

**Proof 5 (Proof of Corollary 1)** *We try to show $R'(f, +1) - R'(f, -1) > R(f, +1) - R(f, -1)$.*

$$R'(f, +1) - R'(f, -1) = \int_{Z'(-1)}^{Z'(+1)} g(x)dx = \underbrace{g(\epsilon')}_{\mathcal{C}} \underbrace{(Z'(+1) - Z'(-1))}_{\mathcal{A}} \quad (43)$$

$$R(f, +1) - R(f, -1) = \int_{Z(-1)}^{Z(+1)} g(x)dx = \underbrace{g(\epsilon)}_{\mathcal{D}} \underbrace{(Z(+1) - Z(-1))}_{\mathcal{B}} \quad (44)$$

*where $Z'(-1) \leq \epsilon' \leq Z'(+1)$ and $Z(-1) \leq \epsilon \leq Z(+1)$ according to Mean Value Theorem. Here $g(\cdot)$ is the density function of standard normal distribution.*

*Next, we try to show: $\frac{R'(f,+1)-R'(f,-1)}{R(f,+1)-R(f,-1)} = \frac{\mathcal{AC}}{\mathcal{BD}} > 1$. Remind that*

$$Z(-1) = \frac{2\sqrt{m+d}\Theta(\gamma)}{\sigma(K^2 - 1)} - K\sqrt{\frac{4m\Theta(\gamma^2)}{(K^2 - 1)^2 \sigma^2} + \frac{2 \log K}{K^2 - 1}} \quad (45)$$

$$= \frac{2\Theta(\gamma)}{(K^2 - 1)\sigma}(\sqrt{m+d} - K\sqrt{m+d+q(K)}) \quad (46)$$

$$\propto (\sqrt{m+d} - K\sqrt{m+d+q(K)}), \quad (47)$$

*where $q(K) = \frac{\sigma^2(K^2-1)\log K}{2\Theta(\gamma^2)}$, and similarly we have*

$$Z(+1) \propto (-K\sqrt{m+d} + \sqrt{m+d+q(K)}), \tag{48}$$

$$Z'(-1) \propto (\sqrt{m} - K\sqrt{m+q(K)}), \tag{49}$$

$$Z'(+1) \propto (-K\sqrt{m} + \sqrt{m+q(K)}). \tag{50}$$

*Therefore, we derive*

$$\frac{\mathcal{A}}{\mathcal{B}} = \frac{Z'(+1) - Z'(-1)}{Z(+1) - Z(-1)} \tag{51}$$

$$\propto \frac{(-K\sqrt{m} + \sqrt{m+q(K)}) - (\sqrt{m} - K\sqrt{m+q(K)})}{(-K\sqrt{m} + \sqrt{m+d+q(K)}) - (\sqrt{m} - K\sqrt{m+d+q(K)})} \tag{52}$$

$$= \frac{\sqrt{m+q(K)} - \sqrt{m}}{\sqrt{m+d+q(K)} - \sqrt{m+d}} \tag{53}$$

$$= \frac{\sqrt{m+d+q(K)} + \sqrt{m+d}}{\sqrt{m+q(K)} + \sqrt{m}} \tag{54}$$

$$= \Theta((\frac{d}{m})^{\frac{1}{2}}) \tag{55}$$

*and*

$$\frac{\mathcal{C}}{\mathcal{D}} = \frac{g(\epsilon')}{g(\epsilon)} \tag{56}$$

*where $Z'(-1) \leq \epsilon' \leq Z'(+1)$ and $Z(-1) \leq \epsilon \leq Z(+1)$*

*Finally,*

$$\frac{\mathcal{A}\mathcal{B}}{\mathcal{C}\mathcal{D}} \geq q(K) \cdot \Theta((\frac{d}{m})^{\frac{1}{2}}) \tag{57}$$

