# OpenReview forum: "To be Robust or to be Fair: Towards Fairness in Adversarial Training"
_ICLR.cc/2021/Conference — Reject_

### Official Review · AnonReviewer4 · 2020-10-29
**Some useful empirical results on class-wise accuracy imbalances**

**Rating:** 5
**Confidence:** 4

**Review:**

This paper begins with the empirical observation that adversarially trained models often exhibit a large different in clean (and robust) accuracies across different classes. This is an important observation, which I have not seen in published work (though I believe is generally understood by many practitioners with the robustness community), and is an important contribution.

The paper then proceeds to use a theoretical example (adapted from the model in Tsipras et al 2018) where adversarial training increases the accuracy difference across classes. More on this later. The paper proposes an algorithm, Fair Robust Learning (FRL), to address this issue. The starting point is a standard Lagrangian-based approach to approximately ensure constraints that the performance for each class should be close to the overall performance. The FRL algorithm then proposes 2 modifications. First, rather than applying the constraint on the robust errors directly, they decompose the robust error into clean error and a robustness term (similar to TRADES) and approximately maintain constraints on both terms. Second, for difficult classes, they propose increasing the adversarial radius, instead of/in addition to only increasing the corresponding Lagrange multiplier. The experiments demonstrate that their FRL approach improves the worst-class clean and robust errors, relative to both standard adversarial training and the baseline Lagrangian approach, without losing much in average-class errors.

Overall, I believe the paper is fairly well-executed and investigates an important topic. The motivating empirical observations are an important contribution. The proposed approach is natural and well-motivated, and the experiments show improvements in the worst-class errors, as should be expected conceptually. The empirical explorations of different variations on the core idea are also valuable.

There were some points in the paper which I believe could be improved.
For me, the theoretical example does not add much insight regarding why this effect occurs.
- Regarding Theorem 1, the comment says that "when the term A is large, the model has close clean errors between the 2 classes, namely $R_{nat}(f, -1) ~ R_{nat}(f, +1)$. Assuming A^2 >> q(K), the terms simplify to:

$$R_{nat}(f, -1) = P[N(0, 1) \leq A(1- \sqrt{K})]$$
$$R_{nat}(f, +1) = P[N(0, 1) \leq A(1- K)]$$

which means we have a $\sqrt{K}$ factor difference between the two errors. In this case, it seems that even before adversarial training, we see a difference in class errors between the classes.

- Additionally, in Theorem 2, using a similar approximation B^2 >> q(K), we'd have

$$R_{nat}(f_{adv}, -1) = P[N(0, 1) \leq B(1- \sqrt{K})]$$
$$R_{nat}(f_{adv}, +1) = P[N(0, 1) \leq B(1- K)]$$

where the two terms differ by a $\sqrt{K}$ factor again, only the leading constant is different.

- The claim in Equation 12 that "the ratio $\Omega$ is large (e.g. >1)" seems to be the result the section builds to, but it seems it is more naturally explained by the fact that if there is already a difference in clean errors, since adversarial training will increase these errors (Tsipras et al 2018), then adversarial training will also scale up the difference in these class-wise errors.

It's possible that this explanation is actually what's happening, and it doesn't seem clearly inconsistent with the empirical data, since in Figure 1 (left), there are already class-wise differences, which generally match the shape of the class-wise differences after adversarial training. But it does seem very different from the story the paper is trying to tell, and it would be good to figure out which one it is (or acknowledge both).

Moving on to the experiments:

- The TRADES baseline has robust accuracy almost 10% lower than the standard TRADES model - can you comment on this?
- Section 4.3 comments "we claim that only upweighting its cost (or Reweight) could not succeed to fulfill the cost-sensitive classification goal in adversarial setting" - could you explain why this is? (after all, reweighting is the basic approach used here too)
- How does the Re-Margin approach compare to the Reweight approach, when the reweighting also involves sweeping over the training $\epsilon$?
- It's nice to see the code. Hopefully this can also be released along with the paper if accepted.

Other notes:

- It would be nice to connect this to the broader fairness literature. This is unfortunately not my area of expertise, but difference in performances between classes (or demographic groups) is a very common metric, and it would be nice to relate this work more closely to work there. "Distributionally Robust Neural Networks for Group Shifts" (Sagawa et al 2020) is a (somewhat arbitrary) example of work which seems related, but I'm sure there are many more. It would be great to connect to this, especially papers which address why/when difference in performance across classes is expected.

I give the paper a 6 overall, though could adjust my rating in either direction depending on the author feedback.


________________

EDIT: Score changed from 6 to 5 during discussion, see comments below.

---

> ### Author Response · Authors · 2020-11-19
> **Response to Official Blind Review 4 -- Part 1**
>
>
> Thank you for taking the time to review our paper.
> We were happy to read that "the observations are important" and "the investigations are well-executed". We are also very appreciated the reviewer's effort to understand our theories. In the following, we will clarify your key concerns.
>
> $\textbf{Question 1}$. In the theory, does adversarial training only just scale up the difference in these class-wise errors?
>
> To answer this question, we did the following further theoretical understandings -- (1) We agree with the point that: in this binary classification setting, there originally exist (negligible) difference between classes errors. (2) Adversarial training will enlarge the disparity but may not just scale up the errors (Z-scores).
>
> $\textbf{(1)}$ Regarding Theorem 1 for clean models, we agree with the reviewer's understanding about our induction. Because the term $A\propto\sqrt{m+d}$ and $A>>q(K)$, so we can have the simplification to get the in-class errors:
> $$R_\text{nat}(f;-1) = \text{Pr.}[\mathcal{N}(0,1)\leq A(1-\sqrt{K})]$$
> $$R_\text{nat}(f;+1) = \text{Pr.}[\mathcal{N}(0,1)\leq A(1-K)]$$
> Thus, there already exist accuracy disparity between 2 classes in a clean model: $R_\text{nat}(f;-1) < R_\text{nat}(f;+1)$.  One thing to note is that: because $K>1$ and $A \propto\sqrt{m+d}$ ($d$ is the dimension of non-robust features and $d$ is large), we can assume these 2 Z-scores are very large negative numbers. Therefore, the errors of both two classes are small and their difference is also small and even negligible.
>
> $\textbf{(2)}$  Regarding Theorem 2 for robust models, because the term $B\propto\sqrt{m}$, where $m$ is the dimension of robust features and much smaller than $d$ (dimension of non-robust features), we cannot have the assumption that $B>>q(K)$. Thus, we cannot directly have the similar simplifications for Eq.(8)(9) as above. As a result, the 2 classes' error disparity condition are not the same between a clean model and a robust model. For a robust model, this ratio between the 2 Z-scores are closely related to the property of the robust model itself, such as the dimension $m$ of robust features it uses. In our main claim in Eq.12, Corollary 1, we clearly stated that adversarial training will enlarge the 2 classes' error difference, given the discussions above that the ratio of Z-scores are not always the same.

---

> > ### Author Response · Authors · 2020-11-19
> > **Response to Official Blind Review 4 -- Part 2**
> >
> >
> > $\textbf{ Question 2}$. The TRADES baseline has robust accuracy almost 10\% lower than the standard TRADES model - can you comment on this?
> >
> > Answer 2: The reason is that the performance in our paper (Figure 1 and Table 3 (upper) in the paper) is based on ResNet18 models while the original paper's reported results of TRADES [Zhang, 2019] are based on Wide-ResNet34. The robust accuracy in ResNet18 is around 10\% lower than the performance of the standard TRADES model in Wide-ResNet34. Table 3 (lower) in appendix shows the performance of TRADES on Wide-ResNet34, which is comparable to the reported performance in [Zhang, 2019].
> >
> >
> > $\textbf{Question 3}$. Section 4.3 comments "we claim that only upweighting its cost (or Reweight) could not succeed to fulfill the cost-sensitive classification goal in adversarial setting" - could you explain why this is? (after all, reweighting is the basic approach used here too)
> >
> > Answer 3. For the reason why only doing reweighting is not optimal, it is because there is originally a trade-off  between the clean error and boundary error [Trades, Zhang 2019] even for samples in one class. Therefore, when only upweighting the boundary errors for a class, it may bring in a side-effect to the increase of this class’s clean error but may not efficiently reduce this class’s boundary error. Thus, it is hard to efficiently balance this trade-off and achieve the fairness constraints in Eq.13. From our experimental results in Figure 3 (left), we see this tension between worst-class clean errors and boundary errors as our debiasing algorithm runs. Although the worst case boundary errors have some tendency to be close to average when we only do reweight, it is not as efficient as doing reweight + remargin.
> >
> > $\textbf{Question 4}$. How does the Re-Margin approach compare to the Reweight approach, when the reweighting also involves sweeping over the training ?
> >
> >
> > Answer 4: From the experimental results in Table 1 in the paper, the Reweight method has the best worst-class clean accuracy and have least sacrifice to the overall clean / robustness performance, but not improve the worst-class boundary errors.  Compared to Reweight, the Remargin method has the effect to better reduce worst-class boundary errors but not effective to balance the clean performance. Reweght+Remargin has the best performance to balance boundary errors and robust errors. However, compared to Reweight, Reweight+Remargin might result in a larger drop on the overall performance.
> >
> >
> >
> > $\textbf{Question 5}$. It would be nice to see the code if the paper gets accepted.
> >
> > Answer 5: We have released our codes at \url{https://drive.google.com/open?id=1IjE11VpV5CGXx633Nef3McgVOoRGMK9x}. We will put it into Github later on.
> >
> > We thank the reviewer's suggestions for further improvements of this work. We will later focus on considering its connections to broader fairness studies.

---

> > > ### Comment · AnonReviewer4 · 2020-11-22
> > > **Response to both author comments**
> > >
> > > Hi,
> > > Thank you for the response. My thoughts inline:
> > >
> > > 1) Theory - Thanks for the clarification. I don't find it fully convincing: you write there is already a $\sqrt{K}$ gap in the clean setting, but that this gap is "negligible" since the accuracy is high in both cases anyways. But then isn't the most straightforward explanation of the observed gap in the adversarial setting that we have a same $\sqrt{K}$ disparity, but since the classification task is harder, it is no longer negligible? Overall, I still stand by my original assessment here from the first comment. I do feel the paper would be stronger without the theoretical results, as they seem to obfuscate more than they explain.
> > >
> > > 2) TRADES - Why not just run WRN-34 results in the main paper? My concern is that the improvements are due to a weak baseline, and the additional table does not include results with the proposed techniques, and does not help to address this concern.
> > >
> > > 3) Thanks - this sentence is still unclear to me: "Therefore, when only upweighting the boundary errors for a class, it may bring in a side-effect to the increase of this class’s clean error but may not efficiently reduce this class’s boundary error." If we upweight the {clean, adversarial} examples of a class, shouldn't that respectively improve performance on {clean, adversarial} examples? (I realize it may not be easy to give a full explanation, and if it turns out the proposed approach just works better empirically, that is fine to say.)
> > >
> > > Overall, I will slightly lower my rating to a 5, as I believe my original concerns have not been addressed (primarily regarding the theory and baselines). Nonetheless, I do believe the initial empirical evaluations are helpful, as are the various explorations into algorithmic variations, and hope the authors continue this work.

---

> > > > ### Author Response · Authors · 2020-11-23
> > > > **Some Discussions about the Theory**
> > > >
> > > >
> > > > Thanks for the response. We agree with the reviewer's opinion. In our paper, we only showed one reason that can explain the "unfair" phenomenon: robust models tends to use fewer features to do prediction, so the classification task is harder and class-wise error disparity is larger.
> > > >
> > > > In addition, we would like to show another reason which causes this "unfair" issue but not related to the "difficulty" level of the classification task. Here, we will show that: when clean models and robust models use same set of features for prediction, robust models will still result the "unfair" issue.
> > > >
> > > > $\textbf{Setting 1.}$ We assume our dataset has the data-label $(x,y)$ sampled from a distribution follows:
> > > > $$ y \stackrel{u.a.r}{\sim} \{-1, +1\},\~\~\~\theta = ( \overbrace{\gamma,...,\gamma}^\text{dim = m})$$
> > > > $$
> > > > x \sim
> > > > \mathcal{N}(\theta, \sigma_{+1}^2I) \~\~\~ \text{if $y= + 1$};\~\~\~
> > > > x \sim\mathcal{N}(-\theta, \sigma_{-1}^2I)\~\~\~ \text{if $y= - 1$},
> > > > $$
> > > > and $\sigma_{+1}:\sigma_{-1} = K$. The only difference between the Setting 1 and Eq.1 in the paper is that: Setting 1 only has one type of features. Next, we will show that a robustly trained model will also introduce "unfair" issue, by reducing the error of class "-1" and increasing the error of class "+1".
> > > >
> > > > $\textit{\textbf{Proof Sketch.}}$ We consider the optimal linear classifiers $f(x) = \text{sign}(\langle \textbf{w},x\rangle+b)$ which minimize clean errors (clean model) and robust errors (robust model). We can simplify the classifier to be $f(x) = \text{sign}(\langle \textbf{1},x\rangle+b/w)$ because both an optimal clean and a robust classifier will give the uniform positive weights $w$ for each feature (shown in Appendix. Lemma 1, we use $\alpha = b/w$ in the following parts). For both models, the clean errors can be formulated by standard normal distribution:
> > > > $$
> > > > R_\text{nat}(f;-1) = \text{Pr.} (\sum_{i=1}^{m}{\mathcal{N}(-\gamma,\sigma^2)}+\alpha\geq0) = \text{Pr.}(\mathcal{N}(0,1)\leq \frac{\alpha - m\gamma}{\sqrt{m}\sigma})
> > > > $$
> > > > $$
> > > > R_\text{nat}(f;+1) = \text{Pr.} (\sum_{i=1}^{m}{\mathcal{N}(\gamma,K^2\sigma^2)}+\alpha\leq0) = \text{Pr.}(\mathcal{N}(0,1)\leq \frac{-\alpha - m\gamma}{\sqrt{m}K\sigma})
> > > > $$
> > > >
> > > > The only difference between a clean model and a robust model is the choice of the interception term $\alpha$. In the following, we will show: the clean trained optimal model $f_{clean}^*$ has smaller optimal interception $\alpha_{clean}^*$ compared to the robust model: $\alpha_{clean}^* > \alpha_{rob}^*$. Then, as a result, the error of class "-1" will drop after adversarial training and the error of "+1" will increase.
> > > >
> > > > $\textbf{(1)}$ For a clean trained model which find an optimal $f_{clean}^*$ to minimize $\frac{1}{2} (R_\text{nat}(f;-1)+R_\text{nat}(f;+1))$ will find the optimal interception: (we denote it as a $g$ function of $\gamma$)
> > > > $$\alpha^*_\text{clean} = \frac{K^2+1}{K^2-1}m\gamma - K\sqrt{\frac{4}{(K^2-1)^2}\cdot m^2\gamma^2+m\sigma^2q(K)}:=g(\gamma)$$
> > > >
> > > > $\textbf{(2)}$ For a robustly trained model, we discuss the case when $\gamma \geq \epsilon$. The optimal interception $\alpha_\text{rob}^*$ is that: (the calculation strictly follows Theorem 2 in the paper)
> > > > $$\alpha_\text{rob}^* = \frac{K^2+1}{K^2-1}m(\gamma - \epsilon) - K\sqrt{\frac{4}{(K^2-1)^2}\cdot m^2(\gamma-\epsilon)^2+m\sigma^2q(K)} = g(\gamma - \epsilon)$$
> > > > By calculating the derivative of $g$ function:
> > > > $$
> > > > \frac{d g(\gamma)}{d\gamma} \geq \frac{K^2+1}{K^2-1}m-K\frac{\frac{4}{(K^2-1)^2}m^2\cdot 2\gamma}{2\sqrt{\frac{4}{(K^2-1)^2}m^2\gamma^2}} = \frac{K-1}{K+1}m>0
> > > > $$
> > > > We can get $g(\cdot)$ is monotone increasing and we have $\alpha_{rob}^*<\alpha_{clean}^*$. Therefore, the clean error of class "+1" will increase but the clean error of class "-1" will decrease after adversarial training. Note that for both clean model and robust model, the error of "-1" is smaller than error of "+1", so the adversarial training tends to results more severe disparity. The reason why this disparity happen can be simply understood as: the robust model's decision boundary ($\langle \textbf{1}, x\rangle + \alpha_\text{rob} = 0$) moves to be closer to the center of class "+1", compared to a clean model.
> > > >
> > > > $\textbf{Theorem 2 in the paper}$
> > > >
> > > > We double checked our inductions in Theorem 2, the true result should be:
> > > > $$
> > > > R_\text{nat}(f_\text{adv};-1) = \text{Pr.}(\mathcal{N}(0,1)\leq B-\sqrt{K\cdot B^2+q(K)} - \frac{\sqrt{m}\epsilon}{\sigma})
> > > > $$
> > > > $$
> > > > R_\text{nat}(f_\text{adv};+1) = \text{Pr.}(\mathcal{N}(0,1)\leq -K\cdot B+\sqrt{B^2+q(K)} - \frac{\sqrt{m}\epsilon}{K\sigma})
> > > > $$
> > > > where we missed the last two terms in the Z-scores. (These 2 errors in the original paper were the training robust errors for the robust models). From the clean errors, we can see the clean error disparity might be due to 2 reasons: (1) the classification task is harder because of the term $B$ is smaller than $A$; (2) Compared to the results in Theorem 1, there is an additional $K$ disparity existing in the clean errors of a robust model.

---

### Official Review · AnonReviewer1 · 2020-10-30

**Rating:** 5
**Confidence:** 4

**Review:**

Thank you for your response, which cleared up some of my questions so I increased my score to a 5. I would have liked more analytical / empirical evidence to substantiate some of the claims made in response to the questions numbered 4, 5, 12, and 13 in the comments provided by the authors. My overall opinion remains unchanged but I encourage the authors to continue this line of work on build on these results and incorporate the feedback provided by the other reviewers as well.

###

This work makes an interesting observation that techniques designed to maximize robustness to adversarial examples may have negative consequences on the model's performance. Typically, this has been studied primarily with an emphasis on accuracy. Here the authors look at fairness instead.

- The first sentence of the introduction makes strong claim about the implications of adversarial examples without providing appropriate references or supporting the claim with evidence.
- Adversarial examples do not have to be "imperceptible", the main requirement is that the perturbation introduced does not affect semantics of the input.
- Adversarial training has been subsumed into verifiable approaches to defending against p-norm bounded adversaries (e.g., randomized smoothing, etc.). Could you comment on why these approaches were left out of the experiments here? They would be able to prove robustness is achieved uniformly for each training example.
- If we know that adversarial training impacts accuracy (as recently re-demonstrated by Tsipras et al.), could you comment on why it would have been expected that fairness would not be impacted given that the training set is balanced among classes and certain pairs of classes are closer to one-another under the p-norm than other pairs of classes?
- The presentation is somewhat confusing given that it entangles the notions of fairness and safety. This seems likely to lead to confusion from the readers. I would recommend splitting the work into two distinct studies: one of the worst-case performance of the classifiers in the absence of an adversary (i.e., safety) and the other being on fairness itself. Unless the authors wish to demonstrate explicitly the connection between safety and certain definitions of fairness, in which case this connection should be presented more upfront in the paper.
- Typo/grammar in "In the lower dimensional space, an optimal linear model is more sensitive to the inherent data distributional difference and be biased when making predictions."
- Could you motivate the choice of dataset here? Why is CIFAR10 a good dataset to study fairness?
- Figure 1: I would recommend using another type of graph. Typically series like this are used to represent data that is sequentially meaningful whereas here a bar plot or something along these lines would probably convey your message better.
- The paper does not explicitly state which fairness definition is used (for instance in Section 2) while this is important here. In particular, the definition considered seems to be based on parity but this is not explicitly stated there. Would the approach considered here apply to other definitions?
- It is not clear what is the applicability of the analysis based on robust/non-robust features is beyond the toy dataset considered here. Could you motivate better why this toy dataset matters in the context you are considering here?
- Grammar error in the statement of Theorem 1
- Grammar in "should not be too larger" (page 5)
- Could you motivate the choice of different thresholds on the natural model error and adversarial model error?
- Missing Section reference on page 6
- Section 4.3 discusses the intricacies and possible tensions between natural and robust accuracies and fairness. The re-margin strategy proposed seems to be at odds with recent results demonstrating that improving robustness to p-norm perturbations conflicts with generalization (see Tramer et al. in ICML20). Have you thought about the suitability of the p-norm constraint placed on adversaries in the context of your approach? Does this create an artificial tension with fairness for the same reason that it creates an artificial tension with accuracy (because the p-norm is not aligned with class semantics).
- Related to this, Table 1 suggests that there is increased tension with natural accuracy when employing FRL compared to the baselines considered. Is that due to the point mentioned above or to what is discussed in Section 4.2?
- The experimental evaluation is limited. It is hard to draw conclusions from one dataset. In particular, I would encourage the authors to consider datasets which are not balanced, which are from other domains than images, and which have a natural motivation for achieving "fairness" due to the task being solved.
- Experiments should also include multiple runs and report variance.
- The discussion of related work is a bit "thin".

---

> ### Author Response · Authors · 2020-11-19
> **Response to Official Blind Review 1 -- Part 1**
>
> Thanks for your time. We were happy to read that "the observations are interesting" and the insightful suggestions to further polish the work. In the following, we will clarify your key concerns.
>
> ........................................................................
>
> Q1. The first sentence of the introduction makes strong claim without providing appropriate references or supporting the claim with evidence.
>
> A1. We add more references and fixed improper claims in Introduction.
>
> ........................................................................
>
> Q2. Missing or broken reference in the paper.
>
> A2. We fixed the broken references in the paper.
>
> ........................................................................
>
>
> Q3. Why verifiable approaches are left out of the experiments?
>
> A3. To investigate the fairness under adv training, we choose to test two most popular and representative algorithms for adv training, i.e., PGD training and TRADES. In addition to these, we further provide experiments on certified defense (Random Smoothing, Cohen 2019) in Table (1*), which also confirms the existence of unfairness issue.
>
> Table (1*)
> | class ( in %) | plane | car | bird | cat | deer | dog | frog | horse | ship | truck |
> |clean acc      |  78.08| 96.15| 78.12 |62.5 | 84.75 |73.53| 92.16 |90.91| 90.91| 90.91|
> |adv acc 0.2   | 64.38| 86.54| 59.38| 37.5 | 62.71 |51.47 |74.51| 72.73 |74.55| 78.18|
> |adv acc 0.4   [  35.62| 63.46 |25. |  10.94 |23.73| 27.94 |47.06| 58.18| 47.27 |69.09 |
>
>
> ........................................................................
>
>
> Q4. If we know that adversarial training impacts accuracy, why it would have been expected that fairness would not be impacted given that the training set is balanced among classes and certain pairs of classes are closer to one-another under the p-norm than other pairs of classes?
>
> A4. This could be one of the reasons why the unfairness issue occurs in some real datasets, but may not be always the case.
>
> In CIFAR10, (see Figure 1 (left)), some classes such as "cat" and "bird" do have 3-5\% lower clean accuracy than average. This may suggest some pairs of classes are indeed "closer" to each other in this model's feature space. However, robust and clean models might use totally different sets of features for prediction [Tsipras 2018, Ilyas 2019]. Thus, a pair of classes which are "close" in the feature space of a clean model might be not "close" or become much "closer" in the feature space of a robust model. Therefore, even though the total accuracy will drop after adversarial training, it is still hard to expect how the accuracy will change in each class.
>
> Moreover, in our theoretical findings in Section 3, we show that the "unfairness" can even exist for binary classification problems where there is only one pair of classes. Thus, this theoretical study demonstrates that the "unfairness" issue can be indeed due to the classification model itself.
>
>
> ........................................................................
>
> Q5. The presentation is somewhat confusing given that it entangles the notions of fairness and safety.
>
> A5. In this work, we focus on the general problem of unfairness in adversarial training. Safety is one of the consequences of such unfairness since certain classes with worst-cast performance can be the attacking target of the adversary.
>
> For example, for autonomous vehicles which apply robust models for traffic sign recognition tasks, if the model is very inaccurate or easy to be attacked only for some specific signs, the vehicle will still be under huge safety risks. In the experiments on the GTSRB dataset, we show the possibility of this “safety” concern under different traffic signs. In a clean traffic-sign classifier trained on GTSRB, we observe that the classifier can achieve high accuracy for every class which is above 92\% (Figure 2 and Table 4 in the paper). However, when we use the adversarial training algorithm, some classes (e.g. "Speed Limit under 100 km/h") become very inaccurate (only has 70\% accuracy) and still very easy to be attacked (only has 16\% adversarial accuracy).
>
> ........................................................................
>
>
> Q6. Typos and grammar error
>
> A6. We have corrected all the typos and grammar errors being pointed out.

---

> > ### Author Response · Authors · 2020-11-19
> > **Response to Official Blind Review 1 -- Part 2**
> >
> >
> > Q7.  Could you motivate the choice of dataset here? Why is CIFAR10 a good dataset to study fairness?
> >
> >
> > A7. The CIFAR10 dataset is a benchmark dataset which is commonly used to study the performance and properties of adversarial training algorithms. In this work, we aim to uncover the general "unfair" property of adversarial training methods from an algorithmic perspective. Thus, many discussions are based on CIFAR10.
> >
> > We also aim to show that our studied "fairness" in image datasets can indeed result in huge realistic consequences. For example, the "fairness" in the image domain can have a huge impact on the safety of computer vision models. In this work, we provide experiments on another dataset (GTSRB) which is composed of 43 classes of images from different traffic signs are showed in Figure 2 and Table 4 in the paper. In a clean traffic-sign classifier trained on GTSRB, we observe that the classifier can achieve high accuracy for every class which is above 92\%. However, when we use the adversarial training algorithm, some classes (e.g. "Speed Limit under 100 km/h") become very inaccurate (only has 50\% accuracy) and still very easy to be attacked (only has 5.6\% adversarial accuracy). This fact shows that the "fairness" in image domain related to adversarial training algorithms is worth great efforts to study.
> >
> >
> > ........................................................................
> >
> > Q9. The paper does not explicitly state which fairness definition is used.
> >
> > A9. In this work, we desire an optimal classification model $f$ which provides equalized clean accuracy and robustness between different classes:
> >
> > 1. Equalized Accuracy: one classifier $f$'s clean error for any class $y\in\mathcal{Y}$ is statistically equal to the model's average clean error: $\text{Pr.}(f(x) \neq y | y = t) = \text{Pr.}(f(x) \neq y)$ for all $y\in \mathcal{Y}$.
> >
> > 2. Equalized Robustness: one classifier $f$'s robust error under certain adversarial attacks for any class $y\in\mathcal{Y}$ is statistically equal to the model's average clean robust error: $\text{Pr.}(\exists~ ||\delta||\leq\epsilon, f(x+\delta) \neq y | y = t) = \text{Pr.}(\exists ~||\delta||\leq\epsilon, f(x+\delta) \neq y)$ for all $y\in \mathcal{Y}$.
> >
> > The "Equalized Accuracy" is well-studied in traditional fairness works which pursue the parity of the model's clean performance for different groups. The "Equalized Robustness" is our new desired "fairness" property for robustly trained models. For every class, the model should also provide equal robustness to resist adversarial attacks. We will update the detailed definitions of our stated "fairness" problem in the paper.
> >
> > ........................................................................
> >
> >
> > Q10. It is not clear what the applicability of the analysis based on robust/non-robust features is beyond the toy dataset considered here. Could you motivate better why this toy dataset matters in the context you are considering here?
> >
> > A10. The toy example is generally used in previous works [Tsipras 2018] to reveal one key behavior of adversarial training: clean models use high dimensional non-robust features for prediction; robust models use low dimensional robust features for prediction. The theoretical investigations based on this setting always reflect the true findings of adversarial training on real datasets [Ilyas 2019, Gourdeau 2019]. Thus, we also inherit it to our work to show the "fairness" issue of adversarial training.
> >
> > In this work, based on the assumption from [Tsipras 2018], we consider a more general condition where the 2 classes have different variances under the mixture Gaussian distribution. From our main theoretical conclusion in Eq. (12), we showed that this variance difference is one key factor to the "unfair" phenomenon of adversarial training.
> >
> > ......................................................................
> >
> > Q11.  Could you motivate the choice of different thresholds on the natural model error and adversarial model error?
> >
> > A11. We desire that for an adversarially trained model, its accuracy disparity among classes should be comparable to disparity level for a clean trained model (which is around 5\%). For boundary error, we choose the thresholds to be 7\% which is a little larger than "clean error" threshold. This is because: for vanilla adversarial trained models, the "boundary errors" have a relatively larger scale than "clean errors" (see Table 1 in the paper). Thus, we intuitively assume that the threshold for "boundary errors" should also be larger than that for "clean errors". In our experiments, when we set the threshold equal to the "clean error" threshold (5\%), we see a moderate drop (5\%-7\%) of the model's overall robustness. In this case, even though the FRL algorithm can still satisfy the fairness conditions, it does not increase the models' worst class robustness. We will add ablation studies about different choices of fairness constraints for further improvements.

---

> > > ### Author Response · Authors · 2020-11-19
> > > **Response to Official Blind Review 1 -- Part 3**
> > >
> > >
> > > Q12. Have you thought about the suitability of the p-norm constraint placed on adversaries in the context of your approach? Does the re-margin method create an artificial tension with fairness?
> > >
> > > A12. As mentioned, Tramer et al. suggested that increasing the adversarial margin might cause the model's clean performance drop. Thus, in our Re-Margin approach, if we increase the margin for a class, it might also decrease the clean accuracy for this class and result in new “unfair” issues for the class-wise clean accuracy. However, we believe that the Re-Margin approach will not cause severe "unfair" issues. The reason is that, during the debiasing process, if there is a class whose clean error exceeds the average by a large margin, FRL algorithm will also upweight the cost of clean errors of that class (Eq.14 and Step.6 in Algorithm 1). Therefore, the clean accuracy of this class can also be guaranteed to be close to average.  In Figure 3, we can see that our FRL (ReMargin or ReMargin+Reweight) can improve the worst-class accuracy and robustness simultaneously which confirms that the Re-Margin approach will not result new unfair issues..
> > >
> > > ........................................................................
> > >
> > > Q13. Table 1 in the paper suggests that there is increased tension with natural accuracy when employing FRL compared to the baselines considered. Is that due to the point mentioned above or to what is discussed in Section 4.2?
> > >
> > > A13. First, it is well known that for many machine learning tasks, debiasing algorithms will decrease the models' overall performance [Agarwal 2018, Zemel 2017, Dutta 2020]. It is usually because: these methods add "fairness" regularizations into the model's training loss and result in a relatively lower accuracy. In this work in adversarial setting, we also have similar observations in our studied fairness problem. For example, when we only apply Reweight strategy (which resembles the most existing debiasing works), we also observed the fact that FRL (Reweight) causes a slight drop of overall accuracy and robustness by 1-2\% (shown in Table 1 in the paper). Thus, the drop of overall performance might be mainly due to our added  “fairness” regularization terms in Eq.14 which is similar to most debiasing algorithms.
> > >
> > >
> > > Second, comparing FRL (Reweight) with FRL (Reweight+Remargin) in Table 1 in the paper, we see a further increase of the model's overall clean error (1.4\%) and robust error (0.1\%). This suggests that the Remargin approach could be another possible reason to cause the tension between fairness and overall performance, in addition to the regularization term in Eq.14. Thanks for the suggested work [Tramer 2020], we will further investigate the trade-off relationship between fairness and overall performance in adversarial setting in the future.
> > >
> > >
> > > ........................................................................
> > >
> > >
> > > Q14. It is hard to draw conclusions from one dataset. In particular, it is suggested to have a natural motivation for achieving "fairness" due to the task being solved. It is also suggested to investigate the unbalanced dataset.
> > >
> > > A.14. We have added one additional dataset, GTSRB to further show the generality of this problem. Under the new dataset, we witness the similar empirical results about the "unfairness" observation (Figure 2 in the paper) and the consistent performance of our FRL debiasing algorithm (Table 2 in the paper).
> > >
> > > References
> > >
> > > 1. A Closer Look at Accuracy vs. Robustness, Yang et al. 2020
> > > 2. Robustness may be at odds with accuracy, Tsipras et al. 2018
> > > 3. On the Hardness of Robust Classification, Gourdeau et al. 2019
> > > 4. Fundamental Tradeoffs between Invariance and Sensitivity to Adversarial Perturbations, Tramer et al. 2020
> > > 6. A reductions approach to fair classification. Agarwal et al 2018
> > > 7. Learning fair representations. Zemel et al, 2017
> > > 8. Is There a Trade-Off Between Fairness and Accuracy? A Perspective Using Mismatched Hypothesis Testing, Dutta et al, 2020
> > > 9. Adversarial Examples Are Not Bugs, They Are Features, Ilyas et al, 2019

---

### Official Review · AnonReviewer3 · 2020-11-02
**A fairness issue in adversarial training**

**Rating:** 6
**Confidence:** 4

**Review:**

Summary:

This paper introduces a fairness perspective on accuracy performance among distinct classes in the context of adversarial training. It makes an observation that adversarial training algorithms (Madry et al. 2017, Zhang et al. 2019) yield biased performances on CIFAR 10. It also offers a theoretical study under a Gaussian mixture setting that respects Eq. (1). Three versions of fair robust algorithms are proposed and evaluated on CIFAR 10.

Strength:

The paper makes an interesting observation on a fairness perspective in adversarial training.
It provides a theoretical insight that tries to explain the observation in a certain setting Eq. (1) in which low-dimensional robust features play a dominant role in adversarial training.
Weakness:

While the observation is new and interesting, this reviewer wonders whether it happens “fundamentally” in any adversarial training setting. The theoretical study definitely offers a deeper understanding, yet it is analyzed under a particular setting Eq. (1) which I am not sure if it represents most practically-relevant scenarios. Also, all the experimental results are provided for the two algorithms (Madry et al. 2017, Zhang et al. 2019) under only one dataset CIFAR 10. As the authors mentioned in Section 6.2, this may be due to particular algorithms and data distribution. If that is the case, it would be good to see that the observation may not happen in other settings. Otherwise (if the observation is fundamental), a thorough theoretical study together with extensive experiments that cover more diverse datasets are desired to be presented.

Theoretical analysis: I can barely follow Theorems 1/2 and their implications stated in the paper. But it was not intuitively clear to me why the unfairness issue occurs in the considered setting. Any elaboration on intuition and insight might help.

There are many grammatical errors and typos.

---

> ### Author Response · Authors · 2020-11-19
> **Response to Official Blind Review 3 -- Part 1**
>
> Thank you for your feedback.
> Your comments of "the observations are interesting'' and "the theory part offers deeper understanding of this topic'' are very encouraging. In the following, we will clarify your key concerns.
>
> Question 1: Whether the “unfairness” phenomenon is “fundamental” for any adversarial training setting? Whether the analysis under setting Eq. (1) is practical?
>
> To answer this question, (1) we first make clarifications on what kind of adversarial training methods that we claim to have this ``unfair'' phenomenon; (2) We provide more empirical evidence to show the existence of the phenomenon; and (3) We explain why our theoretical setting is general and representative for practical-relevant scenarios.
>
> $\textbf{(1)}$ In this work, we concentrate on general adversarial training approaches which follow the robust optimization problem to minimize the average robust error under the same perturbation budgets:
> $$\text{minimize}_f \text{        Pr.}(||\delta||\leq\epsilon, f(x+\delta) \neq y ). $$
> Theoretically, the unfairness issue is analyzed under this adversarial training objective. Empirically, the algorithms being tested, i.e., PGD training and TRADES, are two most popular and representative algorithms for such optimization problem.
>
> $\textbf{(2)}$ In addition, we further provide experiments on one more robust training algorithm in CIFAR10, [Random Smoothing, Cohen 2019], as well as one more dataset (GTSRB). Specifically, Randomized Smoothing is a certified defense which implicitly minimizes model's (certified) robust error. In Table 1*, we obverse the similar "unfair" phenomenon and disparity relation among classes, compared to PGD training and TRADES. In Table 2* (Figure 2 and Table 4 in the paper), we also consider another dataset, GTSRB, which is composed of 43 classes of traffic sign images. In Table (5*), for a clean trained model, most classes have almost 0\% error rates. For an adv trained model, there are several classes that have large errors (e.g. above 20\%) but some classes also maintain 0\% error. Similar observations can be made for class-wise robustness performance in this dataset. All evidence supports that the "unfair" phenomenon is likely to be fundamental for adversarial training.
>
> Table (1*)  Clean / Robust errors of Randomized Smoothing ($\sigma = 0.5$).
> | class ( in %) | plane |  car    |  bird  |   cat  |  deer |    dog |  frog | horse| ship  | truck |
> |   clean acc   |  78.08 | 96.15 | 78.12 | 62.5  | 84.75 |  73.53|92.16 | 90.91 | 90.91| 90.91|
> |  adv acc 0.2 |  64.38 | 86.54 | 59.38 | 37.5  | 62.71 | 51.47 | 74.51| 72.73 | 74.55 | 78.18|
> |  adv acc 0.4 |  35.62 | 63.46 | 25. 0  |10.94 |  23.73| 27.94 | 47.06|  58.18| 47.27 |69.09|
>
> Table (2*) Clean / Robust errors in GTSRB dataset.
> |          (in %)          | Avg. Clean | Avg. Rob. | Best Clean | Best Rob. | Worst Clean | Worst Rob |
> |      clean train.     |       0.5       |      81.1     |       0.0       |       27.5     |        8.3        |      100       |
> |    PGD adv train. |       5.5       |      55.6     |       0.0       |       10.0     |       50.0       |       98.6     |
> |    TRADES train.  |       8.8       |      52.8     |       0.0       |        8.0      |       65.7       |       96.6     |
>
> (3) The theoretical model being analyzed in Eq.1 is generally used in previous works [Tsipras 2018, Ilyas 2019] to reveal one key behavior of adversarial training: robust models use low dimensional robust features for prediction, while non-robust models use high dimensional features. This theoretical setting has been shown to well reflect many properties about adversarial training in real datasets [Ilyas 2019, Gourdeau 2019]. In Eq.1, based on this setting, we consider a more general condition where the 2 classes have different variances under the mixture Gaussian distribution. Thus if the unfairness problem exists for such a simple model, it is likely to exist for more complicated models and data distribution. This assumption is verified in our empirical study.

---

> > ### Author Response · Authors · 2020-11-19
> > **Response to Official Blind Review 3 -- Part 2**
> >
> > $\textbf{Question 2}$. Intuition of the theorems and why unfairness issue could occur.
> >
> > The theoretical analysis was intended to show the unfairness issues could happen generally for adversarial training methods which minimize the average robust error. In the setting of the theoretical model in Eq.(1), we assume the binary class data follows a mixture Gaussian distribution with 2 types of features: robust and non-robust features. This assumption is from works such as [Tsipras 2018, Ilyas 2019], to fundamentally show that: adversarially (robust) trained models only use low-dimensional robust features for prediction; and clean models use high-dimensional non-robust features for prediction. Based on this assumption, we also consider a more general condition where the 2 classes of the binary Gaussian data have different variances (with a ratio $\sigma_{+1}:\sigma_{-1} = K$).
> >
> > In our Theorem 1 and 2, Eqs.(3)(4) compute the 2 classes' standard errors of a clean model (which makes prediction based on high dimensional non-robust features), while Eqs.(8)(9) compute the standard errors of a robust model (based on low dimensional robust features). Corollary 1 is our main conclusion, which shows that: in the high dimensional space (non-robust feature space), the variance ratio $K$ of the 2 classes does not make a big impact so the classes have similar accuracy; while in the lower dimensional space (robust-feature space), the term $K$ becomes a key factor to result in a severe accuracy disparity. Therefore, adversarially trained models will present obvious class-wise accuracy disparity. Similar disparity conditions can also be expected for robustness performance for robust models.
> >
> >
> > References:
> >
> > 1. MMA Training: Direct Input Space Margin Maximization through Adversarial Training, Ding et al. 2018
> > 2.Adversarial Logit Pairing, Kannan et al.2019
> > 3. Robustness May be at Odds with Accuracy, Tsipras et al, 2018
> > 4. Adversarial Examples Are Not Bugs, They Are Features, Ilyas et al, 2019
> > 5. On the Hardness of Robust Classification, Gourdeau et al. 2019

---

### Official Review · AnonReviewer5 · 2020-11-07
**An interesting approach, but there are major concerns**

**Rating:** 5
**Confidence:** 4

**Review:**

The authors study adversarially trained classifiers and observe that the accuracy discrepancy between classes is larger than that of standard models. They then propose a theoretical model where this phenomenon provably arises. Finally, they propose a method to reduce the (standard and robust) accuracy discrepancy between classes, by adapting existing methods from the non-adversarial setting.

I found the paper interesting. The original observation is intriguing and rigorously reproduced on a synthetic dataset. Moreover, the method proposed does seem to improve the class disparity of these models.

However, I have one major concern: it is not clear that the phenomenon observed is a property of adversarial training. A different explanation could be that robust models have lower accuracy than standard ones, with their robust accuracy being even lower. Should the increase in error be multiplicative (which is the most likely scenario) then it would potentially explain the main observation of the paper without taking into account adversarially training at all. Specifically:
- Figure 1: Even the standard model accuracy fluctuates across classes. There are clearly classes that have at least 3 times the error rates of others. The situation for the robust model does not seem particularly different, there are also classes with similar multiplicative ratios in their error, the difference just looks larger because the error rate itself is larger. Measuring the ratio between different error rates might shed some light here.
- Theoretical setting: Looking at equation (12), consider a similar case where the standard error of the robust model is, say, 4 times higher than that of the standard one. In this case, $\Omega$ would be 4 purely based on the accuracy of the model.

To summarize, based on the existing arguments in the paper, we cannot tell apart the scenario where robust training causes class disparity due to an inherent property of the method or simply because it increases the model's error rate. Should the latter be the case, the contribution of the paper would be significantly smaller since the underlying phenomenon would have little to do with adversarial training (similar for the proposed method).

Another issue that was not clear to me is whether the test set is used during training to compute the weighting of the different loss terms (line 5 of algorithm 1). From what I understand, this is the same test set used to report performance, in which case the methodology would be fundamentally flawed and a separate test set (completely unseen) would be needed.

Overall, I do not believe that the paper is ready for publication but I would be willing to update my review based on further discussion.

Other comments (not affecting score):
- Intro: "tradition debiasing..." -> "traditional debiasing..."
- Figure 1: I assume models are trained against an 8/255 adversary, is this correct?
- Eq 14: The fourth error rate should be R_rob, right?
- Section 4.3: broken references
- Section 5: "fairly robust" can be interpreted as "somewhat robust" which is not the intended meaning of "fairly" here

====== POST-RESPONSE UPDATE ======

I appreciate the authors' response and the additional experimental results provided. While I do think that they are a step in the right direction (I hence slightly increased my score to a 5), they still do not address my concerns. Specifically:
- **Empirical results.** Taking a closer look at Table 1* of the response and the results on randomized smoothing (provided in other responses), I do agree that not all errors increase by the exact same multiplicative factor. At the same time, the discrepancy does not seem to be particularly large---i.e., the ratios have a mean of 2.5 with a standard deviation of <0.5. Clearly, given that this is a real-world dataset, it is natural to expect that the effect of adversarial training is not perfectly linear across all classes.
- **Theoretical results.** After reading the response, reading the discussion with Reviewer4, and going through the paper again, I am still not convinced that the increase in class disparity cannot be attributed to a large extent to the gap in clean accuracy introduced by robust training. Perhaps there is a cleaner way of formulating there results or highlighting the key components of the analysis that resolves this. However, unfortunately, I still do not find the analysis convincing in its current state.

Overall, I believe that this point is quite nuanced and the existing empirical and theoretical analysis is not sufficient to draw a confident conclusion. Given how this point is at the core of the paper's contribution, I still recommend rejection.

---

> ### Author Response · Authors · 2020-11-19
> **Response to Official Blind Review 5 -- Part 1**
>
> Thank you for taking the time to review our paper. We were happy to read that "the observations are interesting" and "synthetic analysis is rigorous". In the following, we will clarify your key concerns.
>
> $\textbf{Question 1}$. Whether the observed “unfair” phenomenon is a property of adversarial training?
>
> To investigate this question, we did the following further empirical and theoretical understandings -- (1) we check the clean test performance difference between adversarial training and clean training for each class in CIFAR10; (2) we check the robustness test performance difference between adversarial training and clean training for each class in CIFAR10; (3) we check another dataset (i.e., GTSRB) in addition to CIFAR10; and (4) we further clarify our theoretical analysis.  The evidence from these understandings suggests that adversarial training indeed introduces biased impact on different classes instead of only scaling up the error for each class in a multiplicative way. Next we present the details.
>
> $\textbf{(1)}$ Table (1*) presents the clean test error of clean training and adv training model for each class in CIFAR10, including the corresponding error ratio and difference.  From the results in Table (1*), we can see that in terms of difference (adv - clean), adv training causes the clean error of “bird” and “cat” increase by a large margin, which is around 20\%, much larger than other classes. In terms of ratio (adv : clean), adv training almost triples the errors for class “bird”, "cat" and “deer”, but for class “ship”, it only increases the error by a ratio 1.6. If we take a closer look, for classes "frog" and "ship", they have comparable clean errors in the clean model, but have an obvious discrepancy in a robust model. Thus, the increase of class-wise robust errors are also not equal (in terms of ratio \& difference).
>
> Table (1*) Clean Error in CIFAR10
> |       class ( in %)      | plane | car | bird | cat | deer | dog | frog | horse | ship | truck | avg. |
> |    clean train. err.    |5.6$\pm$0.5 |2.8$\pm$0.4 |9.5$\pm$1.5 | 13.8$\pm$1.3 | 6.1$\pm$1.4 | 10.0$\pm$1.4 | 5.1$\pm$0.9|
> 5.2$\pm$0.9    | 4.4$\pm$0.6 | 4.3$\pm$0.6 | 6.3$\pm$0.7 |
> |     adv. train. err.     |12.9$\pm$1.8|6.8$\pm$0.9|27.9$\pm$2.9|38.4$\pm$2.1|20.8$\pm$1.9|26.9$\pm$3.3|10.2$\pm$1.0|10.9$\pm$1.0|7.1$\pm$1.3|9.4$\pm$1.1| 15.4$\pm$1.5 |
> |   ratio (adv : clean) | 2.3 | 2.4 | 2.9 | 2.9 |  3.4|  2.7 | 2.0 |  2.1 | 1.6 | 2.2 |   2.4 |
> |    diff. (adv - clean) | 7.3 | 4.0 |18.3|22.6|14.6|16.9|  5.1 |  5.7 | 2.7 | 5.2 | 10.3 |
>
> $\textbf{(2)}$  Table (2*) demonstrates the robust error (under PGD attack by 8/255) between clean trained and adversarial trained models for each class in CIFAR10. In terms of adversarial robustness, the disparity between different classes is more obvious. For example, in Table 2* and Figure 1 in the paper, we see for a clean trained model, the adversarial error under PGD attack for each class is around 100\%. However, after adv training, some classes (such as “car”) achieve high robustness with the robust error less than 35\% , but other classes (such as “cat”) still have high robust error which is around 80\%. Thus, the decreases of class-wise robust errors are also not equal (in terms of ratio \& difference).
>
> Table (2*) Robust Error in CIFAR10
> |       class ( in %)      | plane | car | bird | cat | deer | dog | frog | horse | ship | truck |
> | clean train. rob err.|100| 100 | 100 | 100 | 100 | 100 | 100 | 100| 100 |
> |  adv. train. rob err. | 48.4$\pm$2.1| 34.0$\pm$2.5 | 70.4$\pm$3.2 | 82.1$\pm$2.2| 74.6$\pm$3.0| 64.4$\pm$2.7| 50.0$\pm$4.0 | 44.7$\pm$2.3 | 39.5$\pm$4.6 | 43.9$\pm$2.6|
> |   ratio (adv : clean) |  0.48 | 0.34|  0.70 |  0.82  |  0.75 |  0.64  | 0.50 | 0.45 | 0.40 | 4.44|
> |    diff. (adv - clean) | -51.6 | -66. | -29.6 | -17.9 | -25.4 | -35.6 | -50.0| -55.3| -60.5|-56.1|
>
> $\textbf{(3)}$  We also include another dataset (GTSRB, which is composed of 43 classes of different traffic sign images). From the results in Figure 2 and Table 4 in the paper, for a clean trained model, more than 20 classes have almost 0\% error rates and the worst class has 8\% clean error. For an adv trained model, there are 10 classes that have large clean errors over 20\% but some classes maintain 0\% clean error. (Note that the Figure 2 in the paper sorts the classes in descending order for the presentations of both clean and adversarial accuracy.) As a case study, we find that in a clean model, the traffic sign "Speed Limit 60" , "Speed Limit 100" and "Go Straight" signs have 0-1\% clean error, but they have 14\%, 26\%, 33\% error in a robustly trained model. On the contrary, the class "turn right or straight" sign has 3\% clean error in both clean and robust models. It also suggests that the increase of clean error is not proportional between classes. These facts show that the "unfair" condition might be related to adv training and not because the total error is large.

---

> > ### Author Response · Authors · 2020-11-19
> > **Response to Official Blind Review 5 -- Part 2**
> >
> >
> > $\textbf{(4)}$  Theoretically, for example, if the (standard) error of the robust model is 4 times higher than the error of the clean model: $R_\text{nat}(f_\text{adv}) = 4 R_\text{nat}(f)$, it doesn't necessarily imply $R_\text{nat}(f_\text{adv}, +1) = 4 R_\text{nat}(f, +1)$ and $R_\text{nat}(f_\text{adv}, -1) = 4 R_\text{nat}(f,-1)$. In our Theorem 1 and Theorem 2, Eqs.(3)(4) compute the 2 classes' standard errors of a clean model, while Eqs.(8)(9) compute the standard errors of a robust model. By calculating the ratios between 2 classes for both models, we can verify that the 2 classes' errors have different ratios under a clean model and a robust model. Thus, adversarial training does not only scale up the error for each class in a multiplicative way. If we take a closer look, the error ratio between the 2 classes is closely related to the dimension of the features which the classifier uses. For example, this ratio in a clean model is decided by the term $A\propto \sqrt{m+d}$, while the ratio in a robust model is decided by $B\propto \sqrt{m}$, where $m$ and $d$ are the dimensions of robust / non-robust features in the space. Therefore, the unfairness problem is indeed related to the adversarial training method.
> >
> > $\textbf{Question 2}$ : Whether the proposed algorithm uses the test set to adjust training?
> >
> > Answer: During the training process, we use a separated validation set to monitor the robustness / fairness conditions and then adjust our training weights. Therefore, the test samples are totally unseen during the training process. We have revised the description in Alg. 1 to avoid such confusion.
> >
> > Minor Issues:
> >
> > $\textbf{Q.1}$ Intro: "tradition debiasing..." should be "traditional debiasing..."
> >
> > Answer: We corrected the typos in the revision.
> >
> > $\textbf{Q.2}$ Figure 1: I assume models are trained against an 8/255 adversary, is this correct?
> >
> > Answer: In Figure 1, the models are trained against an 8/255 adversary.
> >
> > $\textbf{Q.3}$ Eq 14: The fourth error rate should be $R_\text{rob}$, right?
> >
> > Answer: In Eq. (14), the fourth error rate should be $R_{rob}$.
> >
> > $\textbf{Q.4}$ Section 4.3: broken references
> >
> > Answer: We corrected all broken references in the revision.
> >
> > $\textbf{Q.5}$ Section 5: "fairly robust" can be interpreted as "somewhat robust" which is not the intended meaning of "fairly" here
> >
> > Answer: We modified the improper presentation in Section 5 as the reviewer suggested.

---

### Decision · Program_Chairs · 2021-01-07
**Final Decision**

**Decision:**

Reject

**Comment:**

This paper examines adversarially trained robust models, and finds that accuracy disparity is higher than for standard models. The authors introduce a method they call Fair Robust Learning using Lagrange multipliers to minimize overall robust error while constraining the accuracy discrepancy between classes.

In discussion, consensus was reached that the observations and approach are interesting but the paper is not yet ready for publication. The main concern is that it is not clear if the class accuracy disparity is due to adversarial training, or simply due to lower accuracy in general. Please see reviews and public discussion for further details.